

# An easy-to-use water vapor sampling approach for stable
# isotope analysis using affordable membrane valve multi-
# foil bags
Adrian Dahlmann[1], John D. Marshall[2], David Dubbert[1], Mathias Hoffmann[1] and Maren
Dubbert[1]
[1]Isotope Biogeochemistry and Gas Fluxes, Leibniz Centre for Agricultural Landscape Research (ZALF),
Müncheberg, 15374, Germany
[2]Department of Earth Sciences, University of Gothenburg, Gothenburg, 405 30, Sweden
*Correspondence to*: Adrian Dahlmann (adrian.dahlmann@zalf.de)
**Abstract.** Water-stable isotopes are commonly used in hydrological and ecological research.
Until now, measurements were obtained either by taking a destructive sample in the field (such
as a soil or plant sample) and extracting its water in the laboratory, or by directly measuring it
in the field using semi-permeable membranes. These methods, however, present challenges in
achieving high-resolution measurements across multiple sites since they require significant
effort and resources. Gasbag sampling offers the advantage of non-destructive, cost-efficient,
easy to perform, in-situ measurements without the need to bring a Cavity Ring-Down
Spectroscopy (CRDS) analyzer into the field. Gas permeable membranes (GPM) were utilized
to extract samples of water vapor from the soil, which were then stored in specialized gas bags
(multi-layer foil bags). The bags were tested using laboratory isotope standards for their
maximum storage time, potential memory effects, and reusability. To demonstrate their
applicability in field experiments, in-situ measurements using gas bags were compared to
measurements directly connecting a water stable isotope laser. The storage experiment
demonstrated the ability to store water vapor samples for up to seven days while maintaining
acceptable results for $\delta^2$H and $\delta^{18}$O, although the relative uncertainty was higher for $\delta^{18}$O. A
"Memory experiment" revealed that reusing bags can lead to previous samples influencing
subsequent ones. The experiment on "Combined storage and memory" showed that the duration
of storage increases the effect on memory. The field experiment demonstrated an overall
measurement precision of $0.23 \pm 0.84$ for $\delta^{18}$O [‰] and $0.94 \pm 2.69$ for $\delta_2$H [‰] using the gas
bags. Together, laboratory and field experiments confirmed that the proposed water vapor
sampling system and procedure for stable water isotope analyses using GPM and re-usable gas
bags is a simple, cost-effective, and versatile approach allowing for various applications. We
were able to demonstrate that both 1) storage is possible, and that 2) gas bags can be reused,
since memory effects caused by previous samples can be prevented by appropriate treatment.
This makes the gas bags suited for field collection of water vapor samples for many
applications.



## 1. Introduction

Stable water isotope measurements are used in a variety of scientific fields, particularly in hydrology, ecohydrology, and meteorology, which focus on aspects of the water cycle within the biosphere. The primary isotopes involved are $^{18}O$ and $^2H$ (e.g., Gat 1996; Mook 2001), described as $\delta^{18}O$ and $\delta^2H$ relative to the most abundant isotopes, $^{16}O$ and $^1H$ (Sodemann, 2006). They serve to investigate processes such as infiltration and groundwater recharge (e.g Séraphin et al., 2016), evaporation (e.g. Rothfuss et al., 2010), or the plasticity of root water uptake under stress (e.g. Kühnhammer et al., 2021; Kühnhammer et al., 2023).

Traditionally, the isotopic composition of soil and plant water has been measured through destructive sampling of soil cores or sampled plant material, followed by water extraction e.g. via cryogenic extraction (see method summary Orlowski et al., 2016a) and measured with isotope ratio mass spectrometry (IRMS) analyzers (West et al., 2006; Sprenger et al., 2015). The development of smaller and less expensive cavity ring-down spectroscopy (CRDS) analyzers has led to an increase in potential applications, including e.g. in-situ measurements using gas permeable membranes (Rothfuss et al., 2013; Volkmann and Weiler, 2014; Volkmann et al., 2016; Kübert et al., 2021). Direct measurements are a viable alternative to classic destructive techniques, especially in small plots, as among other benefits (e.g. high frequency measurements) they avoid repeated destructive sampling. However, direct, continuous in-situ field setups are very cost-intensive and technically challenging, requiring a laser spectrometer (e.g. a CRDS) and permanent power supply in the field as well as a strong expertise to maintain. To allow an expansion to a wider set of potential study areas and increase the number of absolute study areas maintainable, scientists are recently trying to develop new simplified sampling systems. This includes capturing soil moisture as water vapor for subsequent laboratory analysis (e.g. Jiménez-Rodríguez et al., 2019; Havranek et al., 2020; Magh et al., 2022; Herbstritt et al. 2023). To do so, primarily glass bottles or gas sampling bags with various fittings are used, which can cost anywhere from less than 50 euros to a couple of hundred euros per container. The advantages of these methods include the ability to quickly measure stored samples in a stable laboratory environment, without the need for time-consuming configuration for specific samples. In addition, multiple sample containers can be filled at once in the field, which allows for the simultaneous measurement of multiple probes, and sampling can generally be performed at a much faster rate. These simplified and more affordable systems could therefore increase the number of studies on stable water isotopes and provide new insights in research.



In this study, we investigated the use of multi-foil bags with septum valves. These bags had
previously been successfully tested for ambient air storage in the laboratory (Jiménez-
Rodríguez et al., 2019). Our investigation focused on exploring the potential of these affordable
bags (< 30€ per bag) for a wider range of applications and particularly for spanning a wide
isotopic range allowing the use in labelling studies. To ensure easy and reliable bag filling and
measurement, we built an additional connection and a portable dry air supply box system for
easy field measurement. We tested the prepared bags in several experiments in the laboratory
using defined standards and, in the field, using comparison to in-situ measurements with a
CRDS. These results allowed us to find a simple approach to using septum-based gas bags for
field measurements of water stable isotopes, which was then tested over a full growing season.
The focus was to investigate storage capability as well as possible isotopic fractionation effects
due to exchange with the inner surface of the bags. Specific objectives included: 1) determining
the maximum storage time of water vapor for accurate measurement of water stable isotopes,
2) testing the reusability of the prepared bags, and 3) confirming these results in a field
experiment. Four different experiments were performed: 1) a storage experiment up to 7 days,
2) a memory experiment with two different standards, 3) a combined storage and memory
experiment, and 4) a field experiment to compare the bag measurements with in-situ CRDS
measurements followed by gas bag measurements over a full cultivation period.



## 2. Material and methods

### 2.1 Study area and basics of stable water isotope measurements

The laboratory experiments were carried out in the laboratories of the Leibniz Centre for Agricultural Landscape Research (ZALF). The field experiments took place at the AgroFlux experimental platform of ZALF, located in the northeast of Germany, near Dedelow in the Uckermark region (N 53°22′45″, E 13°47′11″; ~50-60 m a.s.l.).

During the experiments, the $\delta^2H$ and $\delta^{18}O$ values were recorded using a cavity ring down spectroscopy (CRDS) analyzer (L2130-i, Picarro Inc., Santa Clara, CA, USA). Water vapor from standards and soil samples was transferred to the CRDS analyzer and either measured directly or using the selected gas bags. The hydrogen and oxygen stable isotopes in the sampled water vapor ($\delta^2H$ and $\delta^{18}O$) are detailed in parts per million (‰), relative to the Vienna Standard Mean Ocean Water (VSMOW) through the δ scale (Eq. 1; Craig, 1961).

$$\delta = \left( \frac{R_{sample}}{R_{VSMOW}} - 1 \right) \times 1000 \qquad \text{Eq. 1}$$

The in-situ method used is based on the measurement of water vapor in isotopic equilibrium with the liquid water surrounding the sample probe. To achieve equilibrium between the sampled water vapor and the liquid water, it is imperative to maintain a sufficiently low air flow rate. The flow rate depends on the sample probe length, since the carrier gas needs to be saturated with the sample water. Finally, the isotopic signature between the two phases can then be calculated as a function of the temperature (T) at the phase transition using equations based on Majoube (1961).

### 2.2 Storage and sampling design

The sampling and measurement concept is designed as simply as possible. The storage system is based on multi-layer foil gas sample bags (see table S1 for details) with a membrane-based valve (Multi-Layer Foil Bags, Sense Trading B.V., Netherlands) and an additional self-constructed connector with a valve. The bags have a Water Vapor Transmission Rate (WVTR) of 0.09 g m-2 d-1 (Jiménez-Rodríguez et al., 2019). The connection (Fig. 1) consists of two short PTFE tubes (PTFE-tubing (natural), Wolf-Technik eK, Germany) and an additional luer-lock stopcock (1-way Masterflex™ Stopcocks with Luer Connection, Avantor, USA). A hose clamp (TORRO SGL 5mm, NORMA Group Holding GmbH, Germany) is used to directly connect a ¼-inch tube to the valve and the other 4 mm tube is glued into the ¼ inch tube using 2-component-adhesive (DP8005, 3M Deutschland GmbH, Germany). To protect the adhesive





and ensure proper sealing, electrical isolation tape is wrapped around the splice. Then, a luer-
lock connection (LF-1.5NK-QC, GMPTEC GmbH, Germany) is used to connect the luer-lock
stopcock. The additional connection is necessary to reliably connect the storage system to the
specific experimental setup and to increase reusability.

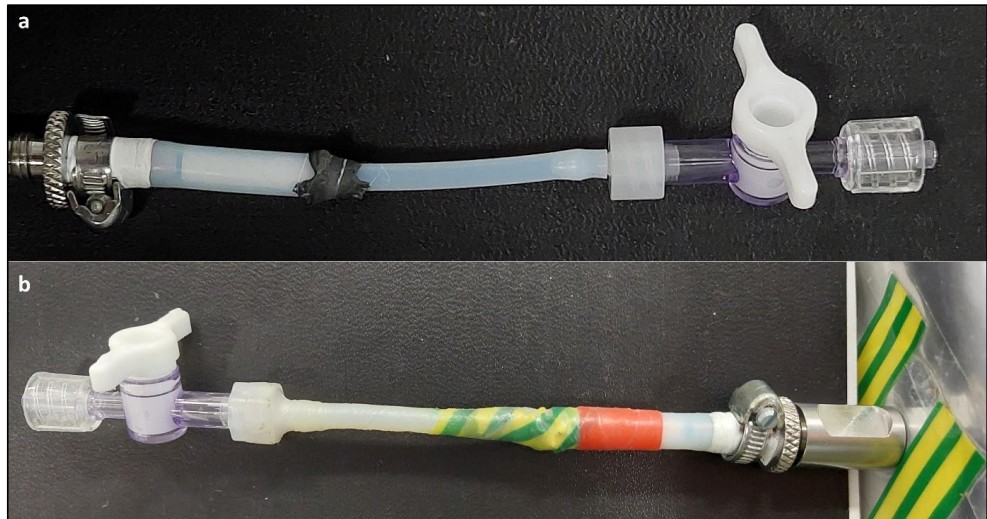

Figure 1: Self-constructed luer-lock connector with the splice exposed (a) and stabilized with tape (b).
During all experiments, water stable isotope signatures ($\delta^2$H and $\delta^{18}$O in ‰) were measured
with the method of Rothfuss et al. (2013), using gas permeable membranes (GPM, Accurel GP
V8/2HF, 3M, Germany; 0.155 cm wall thickness, 0.55 cm i.d., 0.86 cm o.d.). The method has
already been used several times such as in Kübert et al. (2020) or Kühnhammer et al. (2022).
In the laboratory experiments, we attached the GPM to the cap of a 100 ml glass bottle with
two stainless steel fittings (CUA-2, Hy-Lok D Vertriebs GmbH, Germany) to directly measure
standard water vapor and to fill the bags. A gas cylinder was used to induce dry gas at a low
flow rate of 50 - 80 ml per minute. Due to the low flow rate, the water vapor passing through
the GPM reaches an isotopic thermodynamic equilibrium. This means that it has an isotopic
signature that depends on that of the liquid water and the surrounding temperature (Majoube,
1971; Horita and Wesolowski, 1994).
For the 1) direct standard measurements, the sample thus generated is passed directly to the
laser spectrometer to determine its isotopic signature. Since the laser spectrometer only has a
flow rate of approx. 35 to 40 ml per minute, an open outlet was added to ensure a constant flow
and to avoid pressure differences. In addition, the outgoing flow was also measured
continuously, thus ensuring that no ambient air could flow back. A 5-minute average was taken
at the end of a minimum 10-minute measurement for direct standard measurements.





For the 2) field measurements, the GPMs were installed at four different depths (5 cm, 15 cm,
45 cm and 150 cm) and water vapor was transported out of the ground using 4 mm PTFE tubing.
The open ends were fitted with Luer connectors for later connection of gas sample bags and the
dry air supply. To protect these open ends from environmental influences, waterproof outdoor
boxes (outdoor.case type 500, B&W International GmbH, Germany) were installed 20 to 30
cm above the ground. Holes were drilled in the boxes to keep the tubes with cable glands (PG
screw set, reichelt elektronik GmbH, Germany) watertight in the boxes.
A separate box was built to
supply dry air to the
measuring system during
the field experiments (Fig.
2). This contains a pump
(NMP850KPDC-B, KNF
DAC GmbH, Germany)
including a power supply
(DPP50-24, TDK-Lambda
Germany, Germany),
which can transport the dry
air in 3 tubes

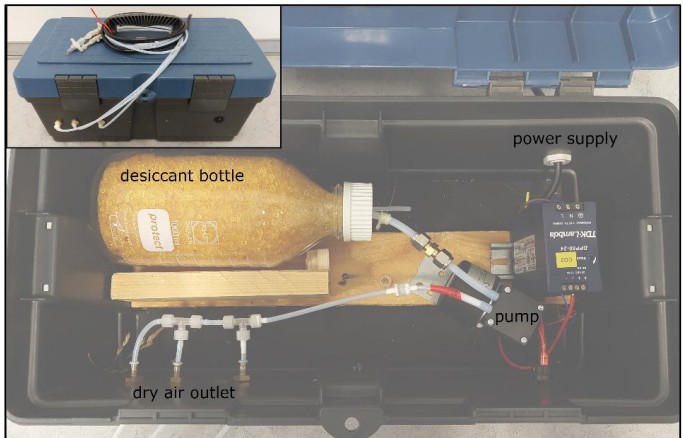

Figure 2: Self-constructed box for field dry air supply (top left) including a bottle with desiccant, power supply and a pump for up to three dry air outlet lines.

simultaneously through the sample tubes (i.e. can fill three gas bags at the same time). The air
is ambient air which is dried by a desiccant (Silica Gel Orange, Carl Roth GmbH + Co. KG,
Germany) contained in a 1-liter bottle (Screw top bottle DURAN®, DWK Life Science, USA).
To regulate the flow of individual sample lines, fixed valves were used (AS1002F-04, SMC
Deutschland GmbH, Germany). As recommended by the manufacturer, care was taken when
filling the bags to ensure that the maximum volume did not exceed 90% of capacity.

**2.3 Laboratory standards**
The water stable isotope measurements were
calibrated against six water vapor standards (see
Table 1) that were manually measured during
the experiments. The standards were each
measured for at least 10 minutes, and a 5-minute
average was documented. Temperature (T) was

Table 1: Standards used during the experiments.

| Standard | $\delta^{18}O_{liquid}$ [‰] | $\delta^{2}H_{liquid}$ [‰] |
|---|---|---|
| L22 | - 19.9 | - 148.1 |
| M22 | - 9 | - 63.3 |
| H22 | 2 | 12.9 |
| L23 | - 16 | - 108.2 |
| M23 | - 9.2 | - 63.9 |
| H23 | - 1.3 | - 32 |

recorded continuously every 30 seconds with a thermometer (EBI 20-TH1, Xylem Analytics



Germany Sales GmbH & Co. KG, Germany) placed directly next to the standard container.
This allowed us to measure the standards in the vapor phase during the laboratory experiments
as well as the later soil samples during the field measurements. Of the six standards with
different δ values, approximately 60 ml were filled into the prepared 100 ml standard bottles as
described in 2.2 (storage and sampling design) and measured directly on the CRDS.

### 2.4 Experimental design: storage test

In our storage experiment, we conducted testing of our gas sample bags for water vapor storage
using water sources of known isotopic composition. New bags, including the self-made
connections underwent initial preparation before being filled with the sample. To eliminate any
production artifacts, each bag was cycled with dry air, filled, and emptied for five times in a
row. Following this preparation, five bags per storage period were filled with two standards,
L22 and M22 (15 min. a 50 ml/min). Throughout the filling process, temperature was
consistently monitored and documented.
Upon filling, the gas bags were promptly measured to ensure that no isotopic fractionation
occurred during the filling process. Subsequently, the gas bags were stored in the laboratory
under stable temperatures (24-25.5°C). Three distinct storage durations - 1 day, 3 days, and 7
days - were chosen before conducting subsequent measurements on the samples. After the
designated storage periods, the samples were measured for 4 to 5 minutes, and a stable 2-minute
average was recorded. To prevent condensation during measurement, the laboratory
temperature was raised to 25°C prior to each assessment.

### 2.5 Experimental design: memory test

Within our memory experiment, we conducted two distinct sample tests, maintaining a
consistent methodology similar to that employed during the storage experiment, utilizing newly
prepared bags.
In the first test, we followed a structured sequence: starting with a direct standard measurement
of the initial standard to ensure carrier gas equilibrium, then filling gas bags with this standard
for subsequent measurements. After emptying the bags, we performed another direct standard
measurement of the initial standard and proceeded to measure the opposite standard. We





repeated the process (fill, measure, empty) with the opposite standard until our measurements
aligned within the required accurate range (defined in 2.8). In the first experiment, L23 was
used as the initial standard and H23 as the opposite standard, in the second experiment, the
standards were used in reverse order. We used five gas bags per standard during the experiments
and the temperature was continuously monitored and documented throughout the filling
process.

### 2.6 Experimental design: combined storage and memory experiment

In the combined storage and memory experiment, we followed a similar procedure to the
memory experiments with one notable difference: after filling the gas bags with the first
standard (L22: -19.9 ‰ $\delta^{18}O$ and -148.1 ‰ $\delta^{2}H$) and conducting measurements, we allowed
the standard to remain in the bags for a one-day storage period and refilled the bags again on
the second day. We then proceeded with the second standard (H22: 2 ‰ $\delta^{18}O$ and 12.9 ‰ $\delta^{2}H$)
following the usual steps until our measurements aligned within the accurate range again.
Between the second and third measurement cycle, the experiment was interrupted due to the
long duration (1h) of each measurement cycle and continued the next day. The bags were
emptied during this second night to avoid any effects. Due to the length of each measurement
cycle, we used 3 repetitions during the experiment and the temperature was consistently
monitored and documented throughout the filling process.

### 2.7 Experimental design: field test

To validate results gained during the laboratory experiments under field conditions, thus testing
the applicability of our proposed system, we compared measurements using the gas bags and
subsequent laboratory analyses with direct in-situ CRDS measurements. The experiment took
place at the area of the AgroFlux sensor platform. We measured once a month during the winter
and once a week starting in the spring resulting in 18 measurement campaigns. During two
measurement campaigns, a total of 50 samples were collected at four different depths: 5cm (n
= 14), 15cm (n = 14), 45cm (n = 7), and 150cm (n = 15). Due to permeability issues, for the
depth of 45cm could only be taken during one measurement campaign, resulting in only 7
samples. For direct CRDS measurements and gas bag sampling, carrier gas was passed through
the GPM soil probe using the described pump system at a flow rate of approximately 50 ml per
minute. First, we connected the CRDS to the outlet valve to determine the time required to
reach a stable value indicating equilibrium. Subsequently, a 2-minute average was recorded for
comparison with the subsequent bag measurement. Second, we connected the bags and filled
them for 15 minutes. The source temperature at the corresponding depth was logged using a
datalogger (CR1000, Campbell Scientific Ltd., Germany) at 20-minute averages.
The field applicability test was followed by gas bag sampling and subsequent stable water
isotope analyses for the same soil depths during a full winter wheat (variety: "Ponticus";
sowing: September 26, 2022; harvest: July 18, 2023) cropping period. We measured once a
month during the winter and once a week starting in the spring resulting in 18 additional
measurement campaigns using only our gas bags. Precipitation was collected within lysimeters
as two-week bulk samples.

**2.8 Calculation of isotope ratios, evaluation of uncertainty and data correction**
The water vapor samples were recorded as 5-minute averages for standards, while bag
measurements were recorded as 2-minute averages, including standard deviation. The isotope
signatures of the collected water vapor water sample were converted to liquid water isotope
signatures using Majoube's method (Majoube, 1971; Kübert et al., 2020). This conversion was
based on the source temperature and assumed thermodynamic equilibrium (Eq. 2 and 3).

$$\delta_{liquid} = (\delta_{vapor} + 1) \times \alpha^+ - 1 \qquad \text{Eq. 2}$$

$$\ln \alpha^+ = a\,\frac{10^6}{T^2} + b\,\frac{10^3}{T} + c \qquad \text{Eq. 3}$$

The equilibrium fractionation factor a+ was determined based on Majoube's (1971)
experimental results, using the coefficients a, b and c (a = 24.844, b = -76.248 and c = 52.612
for $^2H$ and a = 1.137, b = -0.4156 and c = -2.0667 for $^{18}O$).
To assess the accuracy of our measurements, we calculated z-scores for each sample and water
stable isotope ($\delta^2H$ and $\delta^{18}O$). Z-scores indicate the normalized deviation of the extracted water
isotopic ratios from the benchmark isotopic signature of the referenced standard water, and can
be calculated following the method (Eq. 4) described by Wassenaar et al. (2012):




$$z - score = \frac{S - B}{\mu} \qquad \text{Eq. 4}$$

Where S is the isotope signature ($\delta^2$H or $\delta^{18}$O) measured with our gas bag, B is the benchmark
isotope signature and $\mu$ is the target standard deviation. To assess the performance of each
extraction method, we set a target standard deviation (SD) of 2‰ for $\delta^2$H and 0.4‰ for $\delta^{18}$O
for measuring water vapor samples. The target SD was selected based on CRDS measurements
using the bag method and considering standard deviations from previous studies, such as those
by Wassenaar et al. (2012), Orlowski et al. (2016a), and Jiménez-Rodríguez et al. (2019). A z-
score < 2 represents an accurate sample range, a z-score between 2 and 5 describes the
questionable range, and a z-score > 5 representing an unacceptable range (Wassenaar et al.,
2012, Orlowski et al., 2016a, and Jiménez-Rodríguez et al., 2019).



## 3. Results and discussion

### 3.1 Storage experiment

Used laboratory standards, "L22" and "M22", span an isotopic gradient of – 9.0 to - 19.9 ‰ in $\delta^{18}O$ and - 63.3 to - 148.1 ‰ in $\delta^{2}H$ (Fig. 3a; filled symbols: "M22", empty symbols: "L22"). In average, a difference of -0.7 ± 0.6 ‰ $\delta^{18}O$ and -0.1 ± 2 ‰ $\delta^{2}H$ after 1 day, -0.3 ± 0.6 ‰ $\delta^{18}O$ and 4.3 ± 5.2 ‰ $\delta^{2}H$ after 3 days and, 0.4 ± 1 ‰ $\delta^{18}O$ and 0.1 ± 2 ‰ $\delta^{2}H$ after 7 days of storage was obtained for "L22" and "M22". All samples were measured following filling of the bags on day 0 (grey). Except for one sample during the "M22" experiment, deviations from the true standard values in these measurements were all in the range of ± 0.4 for $\delta^{18}O$ and 2 ‰ for $\delta^{2}H$ and thus bias associated with filling of the bags could be excluded.

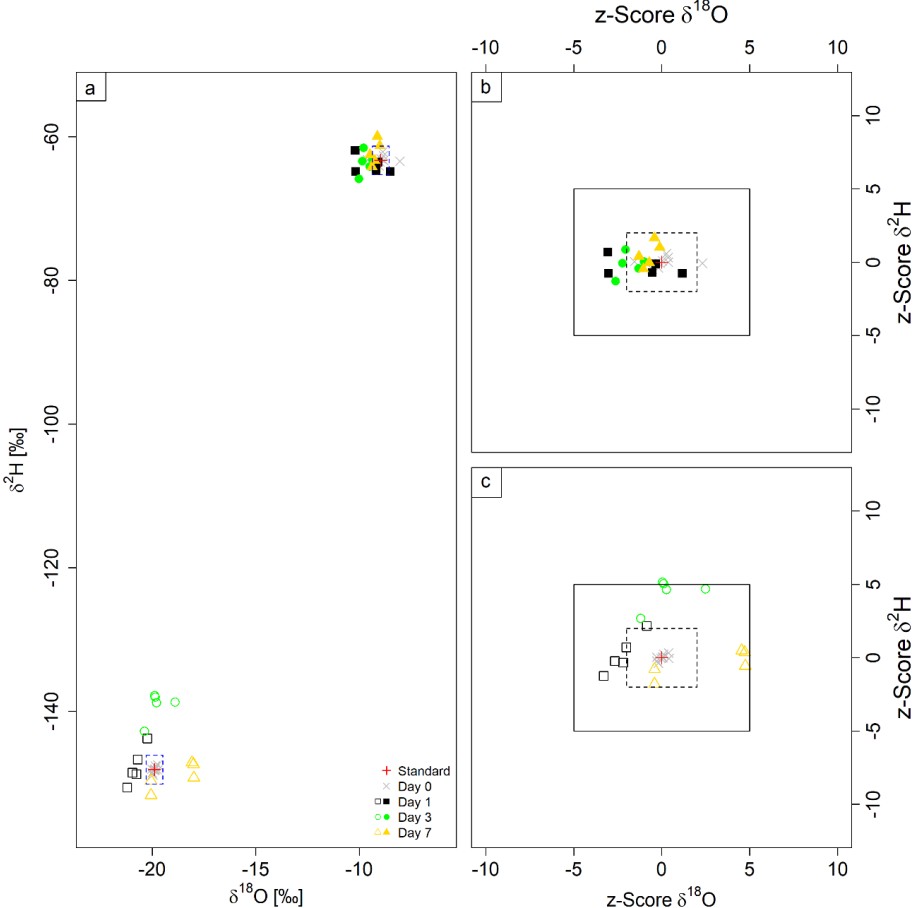

Figure 3: Dual isotope plots showing variation over several days of water-vapor storage in gas bags. The separate panels show results from both experiments (a) and z-score plots for standard "M22" (filled symbols, b) and "L22" (open symbols, c). The black box describes the questionable range while the box delineated with a dashed line describes the accurate range.



All samples were measured following filling of the bags on day 0 (grey). Errors associated with
filling of the bags could be largely ruled out since day 0 measurements were all in the range of
± 0.4 ‰ $\delta^{18}O$ and ± 2 ‰ $\delta^2H$ from the deviation of the true standard values. Only one sample
during the "M22" experiment showed an increased deviation.
The experiment using standard "M22" resulted in an overall high accuracy for all measurements
of the three storage durations with average deviation from the true value (which was - 9 ‰ $\delta^{18}O$
and - 63.3 ‰ $\delta^2H$) being – 0.5 ± 0.5 ‰ for $\delta^{18}O$ and 0 ± 1.6 ‰ for $\delta^2H$. In addition, no trend in
isotopic signature could be observed over storage duration for both $\delta^{18}O$ and $\delta^2H$.
Consequently, z-scores were either within the accurate range or close to it, again with no trend
of decreasing accuracy over storage time.
The second storage test using "L22", showed a higher deviation from the true value (which was
- 19.9 ‰ $\delta^{18}O$ and - 148.1 ‰ $\delta^2H$) being – 0.1 ± 1.1 ‰ for $\delta^{18}O$ and 2.8 ± 4.9 ‰ for $\delta^2H$. No
trend could be observed as in the previous experiment. The increased deviation was mostly
caused by the high imprecision after three days, as all gas bags showed a significant enrichment
(8.9 ± 2 ‰ on average). The higher inaccuracy after three days of storage must be due to an
error during the measurement, as better measurement results were again obtained after 7 days.
The z scores show the same result with accurate values for $\delta^2H$ (except after 3 days) and a larger
scatter with questionable values for $\delta^{18}O$. The average z-score was 0.3 ± 2.7 for $\delta^{18}O$ and 1.4
± 2.5 for $\delta^2H$ (see Table 3 for detailed values).
In comparison to prior studies, testing storage of water vapor samples, our results are generally
of slightly higher accuracy for $\delta^2H$ and comparable for $\delta^{18}O$. The Soil Water Isotope Storage
System (SWISS) introduced by Havranek et al. (2020) showed a high accuracy within the
overall system uncertainty (± 0.5 ‰ $\delta^{18}O$ and ± 2.4 ‰ $\delta^2H$) during a 30-day storage period in
a laboratory experiment. This accuracy is not directly transferable to field experiments, and
several follow up experiments revealed a actual precision of 0.9 ‰ and 3.7‰ for $\delta^{18}O$ and $\delta^2H$
(Havranek et al., 2023). Their system is based on 750 ml glass vials, which are more expensive
and require an offset correction. Magh et al., 2022 developed the VSVS system, which is based
on crimp neck vials in combination with a PTFE/butyl membrane and has a similar accuracy
compared to our results after one day of storage but requires a linear correction for longer
measurement periods. Moreover, although the mean isotopic composition remained the same
throughout the measurement, it increasingly led to very high scatter of the measured isotopic
signatures. Both systems are more difficult to handle compared to inflatable bags as they must





be filled with the same amount of dry gas mixture during the measurement due to the static
properties of the glass vials and the glass vials might also be prone to break during field work.
To the best of our knowledge there are two studies testing different bags for water vapor storage,
and only one using standardized water with different isotopic signatures. Jiménez-Rodríguez et
al. (2019) conducted an experiment in which they filled bags of different material with ambient
laboratory air and measured them after 3 hours, 1 day, 2 days, 9 days, and 16 days. Among the
different bag materials, the MPU gas sample bags – the same bags we used in the present study
- showed the best results with mostly accurate z-scores over the entire measurement period. In
the present study the experiment using standard M22 is best comparable to their result, having
an isotopic signature very similar to the ambient air in our laboratory, yielding comparable
results to Rodriquez et al. (2019) with z-scores in the accurate range. The overall higher scatter
(particularly for $\delta^{18}$O) visible in the experiment using standard L22, which has a different
isotopic signature than the ambient air, led to initial concern over potential exchange with
ambient air. However, we do not think that is likely as the visible scatter already appeared
within one day of storage, was not directed towards isotopic signatures of ambient air and did
not increase over time. We believe the most obvious explanation for this is the previous flushing
with dry air, which was reported by Herbstritt et al. (2023) to lead to an undirected scattering
of the measured values.  This non-directional scattering is more a question of conditioning and
can therefore be attributed to material effects, for example, rather than to an exchange with the
ambient air. Consequently, the memory experiment was performed, to assess potential impacts
of the preconditioning of the bags on the water vapor isotopic measurement results.



## 3.2 Memory experiment

In the first part of the memory experiment (Fig. 4a and b), the initial standard filled into the bags was L23 (-16 ‰ $\delta^{18}O$ and -108.2 ‰ $\delta^2H$), followed by cycles of filling and emptying with standard H23 (-1.3 ‰ $\delta^{18}O$ and -32 ‰ $\delta^2H$). This standard sequence was reversed in the second part of the experiment (initially H23, then cycles of L23). No clear memory effect was found in the first part of the experiment, whereas a clear memory effect was observed in the first repetition (L1) of the second part of the experiment (Fig. 4c), which, however, almost disappeared again in the next repetition (L2). There was an interruption (approx. 45 minutes) between the three measurements with a clear memory effect and the two measurements without a memory effect, so we suspect a connection between storage time and memory effect. The

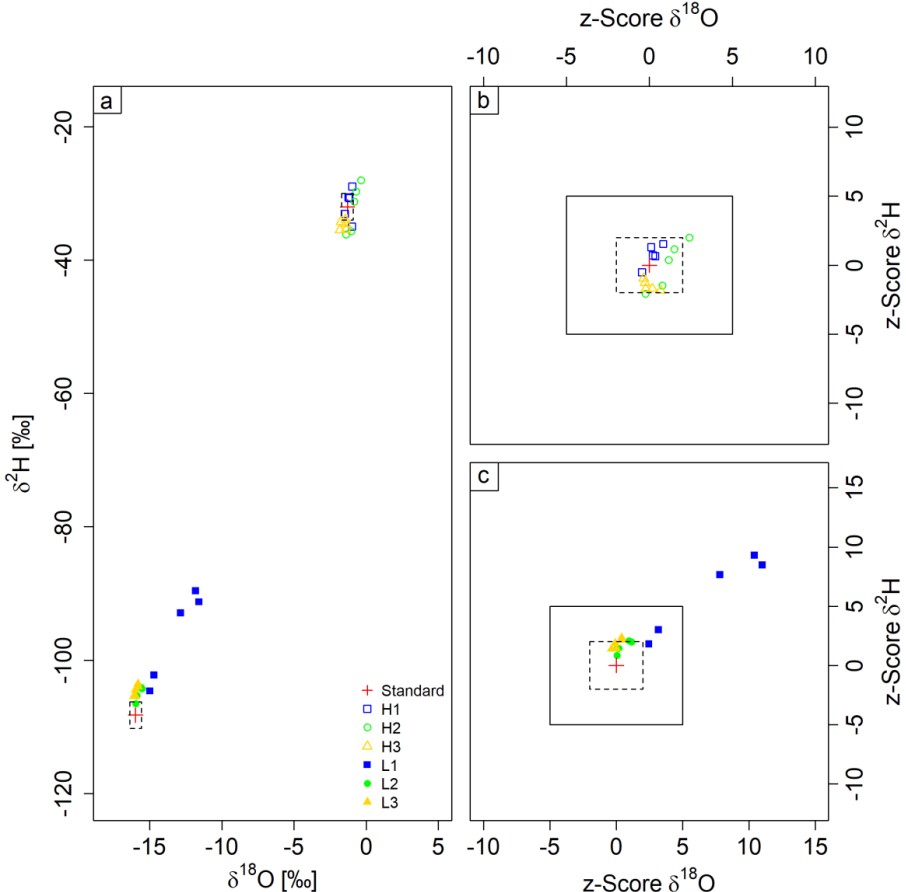

Figure 4: Memory experiment results with dual isotope plot for both experiments (a) and z-score plots for L23 to H23 (b) and H23 to L23 (c). The bags were filled first with standard H, then repeatedly (1-3) with standard L. The memory effect is evident only for measurement L1, the first to follow the change of source water vapor. The black box describes the questionable range while the scatter black box describes the accurate range.



results therefore show that a memory effect caused by the sample previously contained in the
gas bag is possible.
As depicted in Fig. 4 (a and c), except for L1, almost all measurements fall within the standard
deviation for $\delta^{18}O$, while $\delta^{2}H$ values are more scattered around the standard deviation (see table
2). The same pattern can be seen for the z-scores (Fig. 4 b and d). While almost all the z-scores
are in the accurate range or in the questionable range at the threshold of the accurate range, the
values of L1 are clearly outside with values in the unacceptable range. These high z-scores for
L1 are an indication of the memory effect with this first fill. This type of memory effect in the
direction of the last sample contained agrees with the results of Herbstritt et al. (2023). In their
study, the bags were additionally pre-flushed with saturated air of a known isotopic signature.
Some influence in the direction of the water vapor used for rinsing was observed. However,
since we could not detect this effect to a high degree with a traceable direction for a short storage
time in the bag, we performed a combined storage and memory experiment.

**3.3 Combined storage and memory experiment**

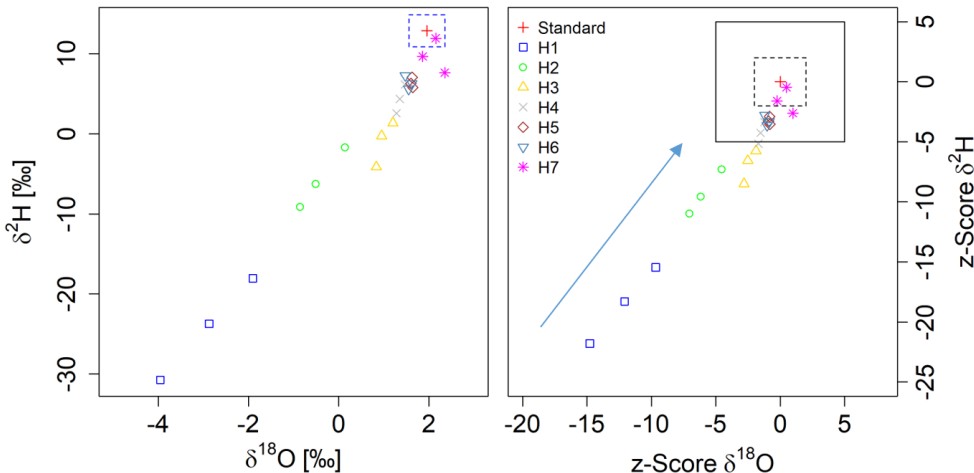

Figure 5: Combined storage and memory effect test with dual isotope plot on the left and z-score plot on the right. The red cross describes the target standard value. The black box describes the questionable range while the scatter black box describes the accurate range. The arrow indicates the direction from strong to weak memory effect.

The final laboratory experiment was conducted as a combined storage and memory effect test.
The bags were stored for 1 day using the initial standard L22 (-19.9 ‰ $\delta^{18}O$; -148.1 ‰ $\delta^{2}H$).
On the second day, the bags were first measured and cycled again with L22 and then with the
opposite standard H22 (2 ‰ $\delta^{18}O$; -12.9‰ $\delta^{2}H$). No significant storage effect was observed



during this one-day storage period, and there was no noticeable difference between the two
repetitions (mean difference between days: $0.4 \pm 0.4$ ‰ $\delta^{18}O$ and $0.1 \pm 1.9$ ‰ $\delta^2H$). However,
when the water source was changed to H, there was a clear memory effect of a magnitude that
has not been described in the literature before (Fig. 5). Measurements H1 to H6 are notably
influenced by the initial standard (table 3). After filling with the opposing standard, H22 (2 ‰
$\delta^{18}O$; 12.9 ‰ $\delta^2H$), the first measurements (H1) revealed a high deviation from the true standard
isotopic value. This high deviation was reduced by around 50% with each repetition until the
average result of H7 is close to the target standard value. The z-scores follow a similar trend
from H1 to H5, gradually decreasing. Although H1 and H2 showed unacceptable z-scores for
$\delta^{18}O$, and H3 fell within the questionable range, all subsequent measurements have z-scores
within the accurate range. The $\delta^2H$ z-scores follow a similar trend to the z-scores for $\delta^{18}O$,
indicating a clear memory effect. However, this effect persisted for a longer duration, requiring
more cycles in the case of $\delta^2H$. The measurements H1 to H3 are in the unacceptable range,
while the results for H4 to H6 are questionable. Accurate values are only observed at H7. On
average, H7 showcase highly accurate results with one measurement at H7 has a z-score within
the questionable range. The transition between the two measurement days, between H2 and H3,
is notably evident in the shift in $\delta^{18}O$ z-scores. The difference of $\delta^2H$ is smaller, but this cannot
be attributed to the overnight break of the measurement, as there is also hardly any difference
between the measurements H4 and H6, which were measured directly one after the other.
However, it is clearly visible that a memory effect is significantly increased by the previous
sample during a longer storage period and remains visible over significantly more fillings.
These results are highly relevant for potential usage of storage bags in especially labelling
experiments. Based on our results, we advise only use the presented method and used bags for
measurements of the natural abundance or samples within the isotopic range of our experiments
or performing additional experiments on labeled water vapor samples. If reused, gas bags
should be repeatedly filled and emptied at least seven times (n≥7) prior to actual sampling.

Table 2: Mean isotopic signature and z-scores of the different repetitions of the combined storage and memory experiment.

| Repetition | Diff. $\delta^{18}O$ [‰] | Diff. $\delta^2H$ [‰] | Z-score $\delta^{18}O$ | Z-score $\delta^2H$ |
|---|---|---|---|---|
| H1 | $-4.9 \pm 1$ | $-37 \pm 6.4$ | $-12.2 \pm 2.6$ | $-18.5 \pm 3.2$ |
| H2 | $-2.4 \pm 0.5$ | $-18.6 \pm 3.7$ | $-5.9 \pm 1.3$ | $-9.3 \pm 1.9$ |
| H3 | $-1 \pm 0.2$ | $-13.9 \pm 2.8$ | $-2.4 \pm 0.5$ | $-6.9 \pm 1.4$ |
| H4 | $-0.6 \pm 0.1$ | $-8.5 \pm 1.8$ | $-1.5 \pm 0.2$ | $-4.3 \pm 0.9$ |
| H5 | $-0.3 \pm 0$ | $-6.5 \pm 0.7$ | $-0.8 \pm 0.1$ | $-3.2 \pm 0.3$ |
| H6 | $-0.4 \pm 0.1$ | $-6.5 \pm 0.9$ | $-1 \pm 0.2$ | $-3.2 \pm 0.4$ |
| H7 | $0.2 \pm 0.3$ | $-3.1 \pm 2.2$ | $0.4 \pm 0.6$ | $-1.6 \pm 1.1$ |

### 3.4 Field test - Comparison between gas bag sampling and direct measurements

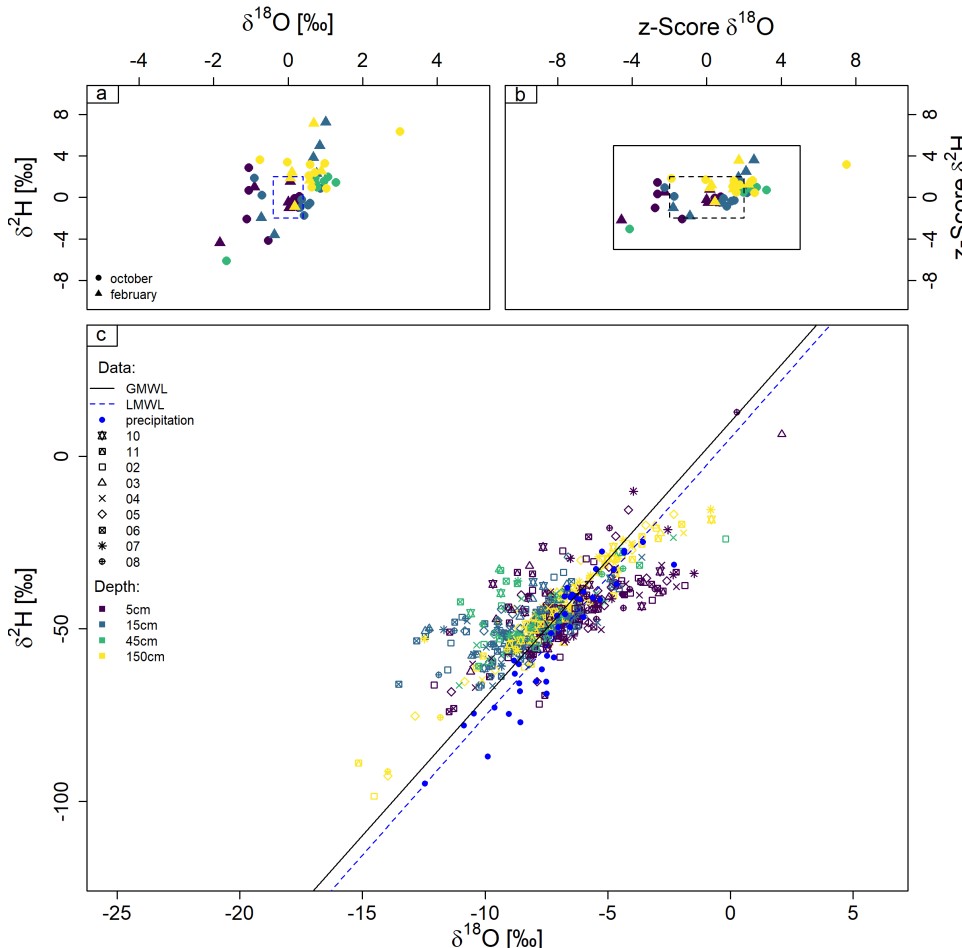

Figure 6: Comparison between in-situ and bag measurements (a) and related z-scores (b). The dual isotope plot (c) shows all 603 measurements taken during the cultivation period. The black box describes the questionable range while the dashed black box describes the accurate range.

To compare the measurements during the two campaigns and calculate the Z-scores, we considered the measured isotopic value of the direct in-situ measurements as the benchmark value (B) and the measurements from the gas bags as the sample (S). Of the 623 measurements taken, 3.2% had to be discarded due to damaged bags, filling errors, or condensation during the measurement and are therefore not shown. To exclude any memory effects, as we saw in the combined experiment for up to seven repetitions, the reused bags were rinsed 10 times.





The average difference between direct measurement and bag measurement was 0.2 ± 0.9 ‰ for
$\delta^{18}O$ and 0.7 ± 2.3 ‰ for $\delta^2H$ during the first sampling campaign in October, 2022 and 0.1 ±
0.8 ‰ for $\delta^{18}O$ and 1.4 ± 3.3 ‰ for $\delta^2H$ for the second sampling campaign in February, 2023
(Fig. 6a). The deviation of the bag method from direct in-situ measurements was thus mostly
within the uncertainty range of the in-situ method and yielded in highly accurate z-scores (Fig.
6b). However, the $\delta^{18}O$ z-scores exhibit a larger scatter compared to $\delta^2H$, consistent with the
results of the laboratory storage experiment. In comparison to other methods determining the
isotopic signature of soil water, the tested gas bag method competed well.  In the past,
destructive
measurements of soil
water have relied
predominantly on
cryogenic vacuum
extraction (CVE). The
accuracy of CVE can
vary greatly for soil
samples, as shown by a
comparative study by
Orlowski et al. (2018), in

Table 3: Absolute measurement values ($\delta^{18}O$ and $\delta^2H$), differences of water stable isotopes (direct vs. bag measurement) and z-scores of the different depth during the two field experiments.

| Depth [cm] | Diff. $\delta^{18}O$ [‰] | Diff. $\delta^2H$ [‰] | Z-score $\delta^{18}O$ | Z-score $\delta^2H$ |
|---|---|---|---|---|
| **25.10.2022** | | | | |
| 5 | - 0.3 ± 0.6 | - 0.6 ± 1.9 | - 0.7 ± 1.6 | - 0.3 ± 1 |
| 15 | 0.2 ± 0.6 | - 0.2 ± 1.1 | 0.5 ± 1.6 | - 0.1 ± 0.6 |
| 45 | 0.6 ± 1 | 0.4 ± 2.9 | 1.4 ± 2.5 | 0.2 ± 1.5 |
| 150 | 0.8 ± 1 | 2.9 ± 1.6 | 1.9 ± 2.5 | 1.5 ± 0.8 |
| **21.02.2023** | | | | |
| 5 | - 0.5 ± 0.8 | - 0.6 ± 2.3 | - 1.3 ± 2.1 | - 0.3 ± 1.2 |
| 15 | 0.4 ± 0.7 | 2.13 ± 4.2 | 0.9 ± 1.8 | 1.1 ± 2.1 |
| 150 | 0.4 ± 0.4 | 2.5 ± 2.6 | 1 ± 0.9 | 1.2 ± 1.3 |

which the results of 16 laboratories showed a mean difference compared to the reference water
ranging from +18.1 to -108.4‰ for $\delta^2H$ and +11.8 to -14.9‰ for $\delta^{18}O$ across all laboratories.
In addition, CVE is associated with co-extraction of organic compounds, significantly
interfering with the isotopic quantification (Orlowski et al., 2018). In comparison, methods
using in-situ soil or xylem probes based on semi permeable tubing have reported high accuracy
(Volkmann and Weiler, 2014; Volkmann et al., 2016; Rothfuss et al., 2013; Kübert et al., 2020).
Among the few previous experiments that tested water vapor storage of soil or plant water in
controlled or field conditions, Herbstritt et al. (2023) sampled prepared sandboxes and achieved
an accuracy of 0.2 ± 0.8 ‰ $\delta^{18}O$ and 0.8 ± 2.9 ‰ $\delta^2H$ after calibration, while Havranek et al.
(2023) achieved an accuracy of ± 0.9 ‰ in $\delta^{18}O$ and ± 3.7 ‰ in $\delta^2H$ during several experiments,
comparable to our findings (0.2 ± 0.9 ‰ for $\delta^{18}O$ and 0.7 ± 2.3 ‰ for $\delta^2H$ in the first sampling
campaign and 0.1 ± 0.8 ‰ for $\delta^{18}O$ and 1.4 ± 3.3 ‰ for $\delta^2H$ in the second sampling campaign).
In the field experiment of Magh et al. (2022), xylem water samples were taken using the
borehole equilibration method (Marshall et al. 2020). In general, the VSVS system did not differ





significantly from the in-situ measured data but resulted in a higher uncertainty with 0.6 ‰ to
0.8 ‰ for $\delta^{18}O$ and 0.6 ‰ to 4.4 ‰ for $\delta^2H$ after.
Measurements of soil water isotope profiles over the full season field experiment (Fig. 6c)
revealed a wide range of isotopic signatures. The isotopic signature of precipitation is
represented by the local meteoric water line (LMWL) shown here for the period of … to ….
The LMWL reveals a slightly different offset but equal increase between $\delta^{18}O$ and $\delta^2H$
compared to the Global Meteoric Water Line (GMWL). The isotopic signature of soil water
can vary strongly from precipitation, as it is a mixture of different precipitation events
containing different isotopic signatures and magnitude. Furthermore, its isotopic signature can
change significantly as evaporated soil vapor is depleted in heavy isotopes, leaving the
remaining soil water enriched in $^{18}O$ and $^2H$ (Dubbert and Werner, 2018). This results in a wide
range of isotopic signatures throughout the complete cultivation season, as can be seen in the
wide scatter around the LMWL. In general, the measurements show isotopic signatures similar
to precipitation immediately after rain events and a trend toward evaporative enrichment in
during droughts. As expected, evaporative enrichment is particularly evident in the upper 5 cm
depth, while there are only slight trends in evapotranspiration enrichment at lower depths (e.g.
Sprenger et al., 2016). These results are consistent with the environmental conditions, as the
measurements were taken during a rather wet cultivation season with only short droughts.
Overall, our findings from the field trial suggest a good agreement with GPM probe and bag-
based soil water isotope measurements with the LMWL and are plausible in terms of seasonal
variability (e.g. compare offsets between cryogenically extracted bulk soil water isotope
measurements and LMWL; e.g. Zhao and Wang, 2021). Notably, there is increased variability
and higher rate of discarded samples at 45 cm depth. This coincides with the placement of the
GPM probes just below the lower boundary of the plow layer. This typically leads to a layer of
increased soil compaction underneath, which we suspect had deteriorating consequences for
the functionality of the GPM probes that should be considered in future experiments in
agricultural settings.
**4. Conclusion**
Our laboratory and field experiments have confirmed that GPM combined with gas bags for in-
situ soil water vapor sampling and subsequent stable water isotope analyses is a reliable, cost-
effective, and easy to handle method allowing for many future applications. We were able to
demonstrate that both 1) storage is possible and 2) memory effects caused by previous samples
can be prevented by appropriate preconditioning, allowing the gas bags to be reused. When



reusing the bags, it was important that 1) the bags were rinsed ten times with dry air, 2) the
additional connection including valve was built and 3) the bags and their valves (especially the
seals) were regularly checked for damage. In addition, great care must be taken to open the bag
valves only minimally for filling and not to fill the bags more than 90% (as specified by the
manufacturer). Regarding the isotopic signature during the experiment, reuse is easier to carry
out with smaller differences between the consecutive samples in the bags, e.g. in the natural
abundance range. However, if a strong labeling experiment is performed, the bags may need to
be handled differently (e.g. better flushing between samples or no reuse). Through the
conducted field experiment, we were able to show that the bags could be used in our case with
an accuracy of $0.23 \pm 0.84$ $\delta^{18}O$ [‰] and $0.94 \pm 2.69$ $\delta^2H$ [‰], which allows a wide
applicability. The possibility to take and store samples easily and without permanent power
supply extends the usability of stable water isotope measurements in the field. Finally, the bags
should not be measured at a temperature that is lower than the temperature measured at the
GPM (source temperature) during the measurement. If the gas bags are measured below the
source temperature, condensation will occur in the bag, which can greatly distort the
measurement result.
**5. Data availability**
The data will be available in the BonaRes repository upon publication.
**6. Author contribution**
AD and MD designed the study. AD conducted the experiments and analyzed the data. JM, DD,
and MH provided support for the experimental setup and analysis methods. AD prepared the
paper with supervision from MD and contributions from all co-authors.
**7. Competing interests**
The authors declare that they have no conflict of interest.
**8. Acknowledgements**
We acknowledge funding by ZALF Leibniz as well as the Leibniz association (ISO-SCALE
project; project number K444/2022). The authors wish to thank the Experimental Infrastructure
Platform (EIP) of ZALF and Linda Röderer for assisting with the field experiments.





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
