# Peer review of "Simple water vapor sampling for stable isotope analysis using affordable valves and bags"

_Atmospheric Measurement Techniques, 2024_

## Referee Comment (RC1)

**Review on manuscript amt-2024-43**

Title: An easy-to-use water vapor sampling approach for stable isotope analysis using affordable membrane valve multi-foil bags

Author(s): Adrian Dahlmann et al.

**General comments:**

In the manuscript by Dahlmann et al. the authors present an alternative approach (in comparison to Herbstritt et al., 2023, Magh et al., 2022 and Havranek et al., 2021) of obtaining water vapor samples for analyzing the isotopic composition of soil water obtained with gas-permeable membranes (GPM) and storing them in multi-layer foil bags. They performed different experiments to test maximum storage time, potential memory effects and reusability as well as field applicability. The authors conclude that their approach is a simple, cost-effective, and versatile approach.
The paper is nicely written and well structured.

However, I have two main concerns:

As approach and concept in this manuscript are very similar to the paper by Herbstritt et al. (2023) - only a different type of GPM and a different (commercially available) bag type were used - the results should be compared point by point and discussed accordingly. Here some revisions and additional considerations are needed.

Moreover, the cited study by Jiménez-Rodríguez is only available as preprint in HESSD. It was under review in 2019 but not accepted due to substantial issues. No revision was provided by the authors afterwards, thus no accepted peer-reviewed version exists. This pre-print can therefore not be cited in your manuscript.

**Specific Comments:**

L. 28: what you call 'precision' is the deviation from the true value, isn't it?

L. 30: (and throughout the manuscript) 'water stable isotopes' not 'stable water isotopes'

L. 34: 'suitable for many applications' was already mentioned in L. 31. Please rephrase

L. 61: 'which can cost anywhere from' sounds a bit sloppy

L. 67: '…provide new insights in research' please be a bit more specific or delete.

L. 98: wrong! ‰ is not "million"

L. 112: P/N of bags? Fitting of bags? Volume of bags?

L. 115: use the manufacturer as reference for material properties such as 'Water Vapor Transmission Rate'

L. 125: Fig. 1: Where is the bag? On the right side in Fig. 1b?

L. 130: how much standard water was added to the 100 mL bottle? And what was the size of the GPM inside the bottle? This is crucial for the isotopic equilibrium

L. 275: Jiménez-Rodríguez can't be cited (see general comments)

L. 289: Fig. 3: would be nice if you could add "L22" and "M22" directly to the figure, there is enough white space. The Legend is also a bit small.

L. 305: if you assume an error during the measurement, I'm wondering why do you present these data and didn't repeat the measurement?

L. 325 and 332: Jiménez-Rodríguez can't be cited (see general comments)

L. 324 to 342: please rephrase and take into account the findings of Herbstritt et al. (2023)

L. 354: Fig. 4: see comments to Fig. 3

L. 378: "...has not been described in the literature before" please rephrase or delete.
I would say it is comparable to Herbstritt et al. (2023), Fig. B1.

L. 378-400: Please discuss and compare the results of Herbstritt et al. (2023) on memory effects and conditioning here.

L. 504 ff: Please check the reference list thoroughly, e.g. Millar et al., (2018) or Orlowski et al., (2016b) are in the reference list but not cited in the text.

**Technical correction:**

L. 11: 'water stable' not 'water-stable'

L. 16: 'easy-to-perform' instead of 'easy to perform'

L. 30: I suggest "…using GPM combined with…" instead of "…using GPM and…"

L. 34: 'suitable' instead of 'suited'

L. 83: please use different bullet points for the field experiments e.g. i) ii) iii) or (a) (b) (c)

L. 98: wrong wording: detailed in

L. 108: Majoube is from 1971

L. 215: 'procedure similar to' instead of 'similar procedure to'

L. 247: please delete line break

L. 284: I suggest 'On average' instead of 'In average'

L. 396: 'especially in' instead of 'in especially'

L. 447: please add dates

---

## Author Comment (AC1)

**Reply to reviewer 1 on manuscript amt-2024-43:**

**Review on manuscript amt-2024-43**

Title: An easy-to-use water vapor sampling approach for stable isotope analysis using affordable membrane valve multi-foil bags

Author(s): Adrian Dahlmann et al.

**General comments:**

In the manuscript by Dahlmann et al. the authors present an alternative approach (in comparison to Herbstritt et al., 2023, Magh et al., 2022 and Havranek et al., 2021) of obtaining water vapor samples for analyzing the isotopic composition of soil water obtained with gas-permeable membranes (GPM) and storing them in multi-layer foil bags. They performed different experiments to test maximum storage time, potential memory effects and reusability as well as field applicability. The authors conclude that their approach is a simple, cost-effective, and versatile approach.

The paper is nicely written and well structured.

However, I have two main concerns:

As approach and concept in this manuscript are very similar to the paper by Herbstritt et al. (2023) - only a different type of GPM and a different (commercially available) bag type were used - the results should be compared point by point and discussed accordingly. Here some revisions and additional considerations are needed.

Moreover, the cited study by Jiménez-Rodríguez is only available as preprint in HESSD. It was under review in 2019 but not accepted due to substantial issues. No revision was provided by the authors afterwards, thus no accepted peer-reviewed version exists. This pre-print can therefore not be cited in your manuscript.

*Thank you very much for the positive feedback on our work. We will revise the manuscript in detail to answer all your questions and those of the second reviewer. As part of this revision, we will remove all statements/references to the preprint by Jiménez-Rodríguez. In addition, as recommended by reviewer 2, we will separate the discussion/results and include a detailed comparison with Herbstritt et al. (2023) and other comparable methods, as well as a detailed recommendation about 1) the use of our method, 2) possible errors and how to avoid them, and 3) what should be tested for future use.*

**Specific Comments:**

1. L28: what you call 'precision' is the deviation from the true value, isn't it?

   *Yes, we will change it to "accuracy" for a better understanding.*

2. L30: (and throughout the manuscript) 'water stable isotopes' not 'stable water isotopes'

   *Will be done.*

3.  L34: 'suitable for many applications' was already mentioned in L. 31. Please rephrase

    *We will change it to: "This makes the gas bags suitable for field collection of water vapor samples for many scientific fields."*

4.  L61: 'which can cost anywhere from' sounds a bit sloppy

    *We will change it to: "... , which can cost from ~1.2 euros to one to two hundred euros per container."*

5.  L67: '…provide new insights in research' please be a bit more specific or delete.

    *Will be done. It now reads: "These simplified and more affordable systems could therefore increase the number of studies on stable water isotopes and provide new insights in research by increasing the number of possible experimental sites and samples."*

6.  L98: wrong! ‰ is not "million"

    *Will be changed.*

7.  L112: P/N of bags? Fitting of bags? Volume of bags?

    *We could not find a part number but we will add a link to the bags and the product name on the website (Multi Foil Bags with Stainless Steel Fitting, https://www.smelltest.eu/en/product/multi-foil-bags-with-stainless-steel-fitting/). These 1liter bags are equipped with a stainless steel 2-in-1 fitting that combines the valve and septum. Simply put, the septum acts as a seal around which air flows out of the sample bag when the valve is open and seals the opening of the sample bag when the valve is closed. We will add this information (and all other we got from the manufacturer) here or in the supplement.*

8.  L115: use the manufacturer as reference for material properties such as 'Water Vapor Transmission Rate'

    *Will be done.*

9.  L125: Fig. 1: Where is the bag? On the right side in Fig. 1b?

    *Yes, the bag is on the right side of Fig. 1b. In Fig. 1a, the connector is disconnected from the bag. We'll explain this better and add another picture with a bag and the connector to this figure.*

10. L130: how much standard water was added to the 100 mL bottle? And what was the size of the GPM inside the bottle? This is crucial for the isotopic equilibrium

    *Thanks for your comment and question. The standard water vapor was generated using a 100 ml glass bottle filled with approx. ~ 60 - 80 ml of standard water. Two semi-permeable membranes (GPM) were placed inside the bottle: 1) one for dry air supply, submerged in the standard water, and 2) one in the headspace for sampling of water vapor sampling and transport to the analyzer. We then continuously passed dry*

*air at a low flow rate (equivalent to flow rates used in common in situ literature) through the water and through the GPM so that the collected vapor was in temperature-dependent water vapor equilibrium with the liquid phase (like e.g. Rothfuss et al. 2013 or Kühnhammer et al., 2021). The measured water vapor concentration was then compared to the saturated water vapor concentration at the given temperature (and pressure) to ensure saturation. The length of the GPM is not as important here as it is more of a safety mechanism to prevent liquid water from entering the tube/analyzer.*

11. L275: Jiménez-Rodríguez can't be cited (see general comments)

   *All statements/references to Jiménez-Rodríguez's paper will be removed from the revised manuscript.*

12. L289: Fig. 3: would be nice if you could add "L22" and "M22" directly to the figure, there is enough white space. The Legend is also a bit small.

   *Will be done.*

13. L305: if you assume an error during the measurement, I'm wondering why do you present these data and didn't repeat the measurement?

   *We did not repeat the measurement mainly because the results after 7 days looked promising and supported our reasoning against a potential storage effect. Following recommendations of reviewer 2, we will split the results and discussion and discuss them in more detail in the revised manuscript.*

14. L325 and L332: Jiménez-Rodríguez can't be cited (see general comments)

   *All statements/references to Jiménez-Rodríguez's paper will be removed from the revised manuscript.*

15. L324 to L342: please rephrase and take into account the findings of Herbstritt et al. (2023)

   *In the discussion of the revised manuscript, we will include a detailed comparison with Herbstritt et al. (2023).*

16. L354: Fig. 4: see comments to Fig. 3

   *Will be done.*

17. L378: "...has not been described in the literature before" please rephrase or delete. I would say it is comparable to Herbstritt et al. (2023), Fig. B1.

   *I think you are referring to Fig. 5b with the SDs of isotope readings from vapor sampling bags, stepwise conditioned with dry synthetic air. A comparison will be included in the revised discussion.*

18. L378-400: Please discuss and compare the results of Herbstritt et al. (2023) on memory effects and conditioning here.

   *Will be done as part of the revised discussion.*

19. L504 ff: Please check the reference list thoroughly, e.g. Millar et al., (2018) or Orlowski et al., (2016b) are in the reference list but not cited in the text.

   *Thank you very much. We will check all references after all changes have been incorporated.*

**Technical correction:**

1. L11: 'water stable' not 'water-stable'

   *Will be done.*

2. L16: 'easy-to-perform' instead of 'easy to perform'

   *Will be done.*

3. L30: I suggest "…using GPM combined with…" instead of "…using GPM and…"

   *Will be done.*

4. L34: 'suitable' instead of 'suited'

   *Will be done.*

5. L83: please use different bullet points for the field experiments e.g. i) ii) iii) or (a) (b) (c)

   *Will be done.*

6. L98: wrong wording: detailed in

   *Will be done.*

7. L108: Majoube is from 1971

   *Will be done.*

8. L215: 'procedure similar to' instead of 'similar procedure to'

   *Will be done.*

9. L247: please delete line break

   *Will be done.*

10. L284: I suggest 'On average' instead of 'In average'

    *Will be done.*

11. L396: 'especially in' instead of 'in especially'

    *Will be done.*

12. L447: please add dates

    *Will be done.*

---

## Author Comment (AC3)

**Reply to reviewer 2 on manuscript amt-2024-43:**

**R2: General comments**

The authors present a new method of collecting discrete vapor samples for water vapor stable isotope analysis using inflatable multi-foil bags. The presented method contributes to a new, currently evolving field of stable isotope analysis still lacking an agreed-upon standard procedure suitable and attractive for many users interested in performing in situ isotope assays without field-access to an analyzer. Therefore, any reported experience in this regard is highly welcome and I recommend publication after proper revision.

The manuscript describes the use of bags, which differ only in valves (which do not seem to have an effect) from the ones used in a previous study (Herbstritt et al., 2023, doi: 10.5194/hess-27-3701-2023). I therefore suggest a more thorough discussion emphasizing how this work expands the findings of the previous study. Moreover, I don't understand how the proposed treatment of previously used bags would help to get meaningful results if reused for unknown samples. I have a feeling that the tested treatment to remove memory effects does not account for the potential conditions faced by researchers interested in using the proposed method regarding, e.g., feasible or necessary storage time and range of previously observed isotope values.

Formally, the authors decided to combine results and discussion. Unfortunately, this often leads to a limited description of the results. I believe the manuscript would benefit from a better distinction between description and interpretation of the presented findings. Also, some additional technical details (flow measurement devices, new or reused bags for the field test, rinsing atmosphere prior to reuse, etc.) should be added to the method section. Finally, a detailed SOP listing suggested settings and potential pitfalls may be helpful for future users of the proposed method.

*Thank you very much for your detailed and constructive comments and for recommending publication. We will thoroughly revise the manuscript to explain in detail all questions raised in the comments or to clarify misunderstandings. In particular, we will highlight the differences between our work compared to previous studies more clearly:*

*In situ measurements of water stable isotopes are usually performed with two different systems: the recently commercially available WIP system (as used in the study by Herbstritt et al., 2023) and originally developed by Volkmann et al., (2014; for soils) and (2016, for xylem of trees), and home-built systems with GPMs (following the original developments of Rothfuss et al., 2013 and as used in Kübert et al., 2020 or Kühnhammer et al., 2021). The main difference between these systems is that the WIP system dilutes the sample flow by reducing the water vapor concentration in the probe, hence enabling measurements with relatively constant water vapor concentrations. Home-built systems with GPMs usually measure the saturated airflow without dilution in the GPM. One of the main differences is that the water vapor concentration of a sample from the WIP system is usually lower than that of the self-built systems due to the dilution. This has the advantage of reducing the risk of condensation, but also leads to a lower water concentration and thus a reduction in sample volume. We believe that aside from the material used for the storage container (different types of bags, glass vials etc.), the in situ method itself is also an important part that can influence the method development of a new storage method and find it relevant to compare our approach using a self-constructed in situ system with the WIP system used in Herbstritt et al. 2023. We will elaborate more clearly on this and all other important differences between our work and that of Herbstritt et al, (2023) during the revision of the manuscript.*

*Moreover, we agree that rinsing 10 times with dry air is not completely transferable, but our recommendation was based more on our results from the field experiment in February, where we*

*followed exactly this principle (rinse used bags from October 10 times). However, we see that this is not fully explained in the current version of the manuscript and that the difference in the isotopic signal of the samples is also not nearly as strong as for the two standards (see 3.4 Field test), which limits the recommendation. For this reason, we will 1) explain our field experiments in more detail to avoid misunderstanding and 2) perform an additional experiment following our field protocol in which we will store one standard in new bags for one day, rinse the bags with dry air, and then fill them with the opposite standard. We will then measure these samples one (and 3) days later and additionally present and discuss these results in the updated manuscript. (Unfortunately, we were not able to perform and present this test immediately for this reply, as we do not have all the necessary materials in stock). This will give more insights into the reusability of our method under different experimental settings (e.g. natural abundance vs. labelling approaches).*

*As recommended by the reviewer, we will split the Results and Discussion sections to make the results more understandable and to discuss them in more detail. In the discussion, we will include all information requested by the reviewers and compare our results with those of other studies (especially Herbstitt et al., 2023, but also Magh et al., 2022 and Havraneck et al., 2020). We will be more explicit about the differences (e.g. home-made GPM vs. WIP system or glass bottles vs. bags or lab vs. field experiments or different storage times) and point out important advantages and disadvantages in a more comparative way. In addition to these comparisons, we will add a section on how to use our system (SOP) and how to avoid potential problems.*

I provide a list of specific comments below.

**Specific comments**

L10: "water stable", not "water-stable"

*Will be changed.*

L16: "easy-to-perform, in situ", not "easy to perform, in-situ" (also elsewhere in the MS: "in situ" without hyphen)

*Will be changed.*

L22: insert "spectrometer" or equivalent after "laser"

*Will be changed.*

L25: "can lead" seems too weak, as there will always be influence of previous samples. I suggest "does lead" or "will lead"

*Will be changed: "will lead"*

L26: Consider rephrasing to: "…showed that the memory effect increases with duration of storage."

*Will be changed.*

L28: You state the precision, which describes the scattering of repeated or replicate measurements. What is the accuracy, i.e. the deviation from the target value?

*Thank you very much. We will add additional text on the accuracy so this wrong wording will be changed.*

L30 (and elsewhere): "Water stable", not "stable water"

*Will be changed.*

L38f: I do not see why hydrology and meteorology would focus on the biosphere. Consider rephrasing.

*Will be changed.*

L59 (and elsewhere): Do not cite preprints like Jimenez-Rodriguez et al. (2019). It is against AMT guidelines and it devalues the work of reviewers. Especially, do not call such work "successful" (L70) when in fact it has not successfully passed a peer-review process.

*All statements/references to Jiménez-Rodríguez's paper will be removed from the revised manuscript.*

L61: "less than 50 euros" is quite vague. Can you be more specific?

*We will change it to: "... , which can cost from ~1.2 euros to one to two hundred euros per container."*

L63: in what aspect is the lab environment stable? – Temperature?

*Yes. We will change it to "temperature-stable laboratory environment".*

L63: What do you mean by "configuration"?

*This can be misleading due to our wording. By "time-consuming configuration" we meant the time-consuming post-procession step (calculation) to obtain the true isotopic value. Using the glass bottle method requires the filling of missing sample volume with dry air to avoid suction inside the glass bottle. Naturally, this involves an additional correction step, due to the vapor concentration significantly declining over the measurement period (see section 2.1 VSVS lab test in Magh et al., 2022).*

*In our opinion and after testing the glass bottle method ourselves in collaboration with part of the co-author team of the Magh et al., paper, we concluded, that a storage method based on bags rather than glass vials is easier to handle. With the gas bag method, isotopic signatures of the gas inside the bag are directly recorded with a stable water vapor concentration.*

*Of course, this does not necessarily imply that one method is superior to the other, hence we clarify and present method comparisons in a more balanced way.*

L72: Again, be more specific about pricing. This helps other researchers considering using your method. In addition, the numbers never appeared in the manuscript again, i.e. they were not discussed. Nonetheless, you refer to them prominently in the manuscript's title. How do they compare to the per-sample costs of the containers used by Magh et al., (2022, doi: 10.5194/hess-26-3573-2022) and Herbstritt et al. (2023)?

*We will include detailed prices (as an overview table) in the revised version and compare them with Magh et al. (2022) and Herbstritt et al. (2023) in more detail within a new part of the discussion.*

L95f: Please be more specific. How was vapor from standards produced? Was it in equilibrium with the liquid phase (resulting in temperature-dependent isotope fractionation) or flash-evaporated (with no fractionation)?

*Thanks for your comment and question. The standard water vapor was generated using a 100 ml glass bottle filled with approx. ~ 60 - 80 ml of standard water. Two semi-permeable membranes (GPM) were placed inside the bottle: 1) one for dry air supply, submerged in the standard water, and 2) one in the headspace for sampling of water vapor sampling and transport to the analyzer. Both GPMs were sealed with adhesive. We then continuously passed dry air at a low flow rate (equivalent to flow rates used in common in situ literature) through the water and through the GPM so that the collected vapor was in temperature-dependent water vapor equilibrium with the liquid phase (like e.g. Rothfuss et al. 2013 or Kühnhammer et al., 2021). The measured water vapor concentration was then compared to the saturated water vapor concentration at the given temperature (and pressure) to ensure saturation.*

*We will explain this in more detail with corresponding references in the revised manuscript.*

L98: per mil, not parts per million (I wonder how this went unnoticed by five co-authors…)

*Will be changed.*

L108: The Majoube paper is from 1971, not 1961.

*Will be changed.*

L112f: Would it be possible to state a part number for these bags as well? I am unable to find this product in a web query. In addition, how does a membrane-based valve work? Does the sample have to pass through a membrane?

*We could not find a part number to find it on the website but we will include all information we have about the bags in a updated table in the supplement where we can additionally add a link to the bags and the product name on the website (Multi Foil Bags with Stainless Steel Fitting, https://www.smelltest.eu/en/product/multi-foil-bags-with-stainless-steel-fitting/).*

*These multi-foil bags are equipped with a patented 2-in-1 stainless steel fitting. This fitting combines the valve and the septum in one. Simply put, the septum acts as a seal around which air flows out of the sample bag when the valve is open and seals the opening of the sample bag when the valve is closed.*

L115: This number seems to be huge! Assuming that the sample bags (front and back) have an area of roughly one tenth of a square meter, more than half of a sample (which comprises ~17μL or 17mg of water per 1 L air at room temperature when saturated) would be exchanged per day. Can this be true? Please, also state the conditions (temperature, relative humidity, vapor pressure gradient), under which the water vapor transmission rate was determined (without citing a preprint). Otherwise, this number is meaningless. Or disturbingly high.

*Thank you for your comment. In fact, this number is not correct. We calculated the WVTR again and the correct value is 0.00465 gr/m2/24h. Here is the manufacturer's information and the calculation in metric system:*

*Water vapor transmission rate (FED 101): < 0.0003 gr / 100 in² / 24 hrs*

- *100 in² = 0,064516 m² → 1550 in² = 1 m²*
- *< 0.0003 gr / 100 in² / 24 hrs * 15.5 = < 0.00465 gr / m² / 24h*

*With a bag area of ~ 640 cm² it would be:*

- *< 0.00465 gr / m² / 24h * 0.064 = 0.0002976 gr / bag area / 24h or*
- *< 0.2976 mg / 24h for a bag.*

*With 15.3 mg of water sample in 0.9 L of air at room air temperature at saturation, this would be ~ 2% per day or ~ 14 % per week, but (as you already mentioned in your second question about the conditions) this is an extreme value tested with the "FED-STD-101 – Test Procedure for Packaging Materials" at high water concentrations (90%) on one side and low water concentrations (desiccant) on the other side at ~ 38°C ([http://www.woodencrates.org/standards/FED-STD-101.pdf](http://www.woodencrates.org/standards/FED-STD-101.pdf)).*

*We will explain/discuss this in more detail in the updated manuscript.*

L121: Did you test a version without electrical isolation tape that did not work? I am wondering if the tape really makes a difference regarding proper sealing.

*It is true that the electrical tape per se is not important for proper sealing. Initially, we tested the bags without tape, but the adhesive in combination with the PTFE tubing can break under tension, which (of course) leads to leakage. Therefore, we used the electrical tape to stabilize the connector (you could probably use any tape, but we had the electrical tape in abundant stock). We will explain this in the revised manuscript.*

L127: What was the length of the GPM?

*The length of the GPM is not as important here (< 5 cm), as the dry air passes the standard water, and it is more of a safety mechanism to prevent liquid water from entering the tube/analyzer. In the field experiment, we used approx. 12 cm GPM (comparable to soil GPM in e.g. Kühnhammer et al., 2021).*

*For further details, see comment on L95f.*

L133: How was the flow rate measured? And what would have been the maximum possible flow ensuring equilibrium given the GPM length you selected?

*The flow was measured with a RS PRO air flow sensor (257-6409, RS Components GmbH, Germany). Here, we are talking about flow rates during our laboratory experiments with nearly unlimited water supply within the standard botte. We tested the standard bottles used starting with the minimum flow the picarro needs to operate (around 35 ml / min) and increased the flow up to 300 ml / min. Until around 100 ml / min (75 ml / min + picarro flow), it resulted in accurate results. With 100 ml / min + picarro flow the water concentration started to decrease slightly (with still acceptable results). A higher flow rate of 150, 200 and 300 ml/min + picarro flow then resulted a depletion of heavy $^{18}O$ and $^{2}H$ isotopes relative to the standard.*

L135: Under non-EQ conditions, the vapor isotopic composition would also depend on water isotopic composition and surrounding temperature. But not exclusively.

*This is true, we will explain/discuss this in more detail in the updated manuscript. At equilibrium, the estimation of liquid isotopic composition is particularly straightforward, but we will also mention conditions under non-equilibrium conditions.*

L140: By "outgoing", do you mean the flow going out of the sample vessel or the flow going out of the open outlet?

*We are talking about the "open split". We changed it for a better understanding:*

*"Since the laser spectrometer only has a flow rate of approx. 35 to 40 ml per minute, an open split was added to ensure a constant flow and to avoid pressure differences. The open split was continuously measured to ensure that no ambient air could flow back."*

L163: How dry was the air after passage through the desiccant? Was this value tested and constant over the course of the experiment?

*Prior to our experiments, we measured the outlet concentration of the dry box over the course of one day. During the experiments, we regularly tested the water concentration before and after the field campaigns and could not detect any increase after one day in the field. The water concentration of the dry air produced was about 200 ppm. However, the use of such a system should always be tested for the specific application, as a very high flow rate combined with very humid air could greatly affect the duration of possible use.*

L166f: What would happen, if the bags were filled to more than 90% capacity? And why isn't a lower filling capacity stated in the first place? How about filling only to the minimum volume necessary to reach a plateau on the analyzer during analysis? Did you play with that variable as well? How would that impact feasible sample throughput? How would the reduced sample volume affect its vulnerability, e.g., regarding memory effects?

*Thank you for these interesting questions. We will discuss it in more detail in the discussion section, but to answer them:*

*Overfilling can lead to damage to the bags and probably to a much higher stress on the material. At the beginning of our tests, for example, we found that the bags showed folds/creases after being overfilled, which were then repeatedly creased in the same way, leading to material damage. This is indeed very important especially when it comes to reusing the bags, so we now mention this in the updated manuscript at the beginning of chapter 2.2.*

*Personally, we do not recommend a lower filling quantity, as this could change the volume to area ratio and increase the effect of the water vapor transmission rate. In turn, this could potentially increase storage and memory effects.*

*A reduced sample volume could potentially have a positive effect on sample throughput in the field, as the filling time would be significantly reduced. However, a higher sample throughput could also be achieved by simply using multiple dry air pumps, i.e. filling the bags simultaneously in the field, without having to reduce the sample volume.*

L173ff: This statement is a repetition of L141f. Consider deleting.

*Will be changed.*

L180: 100 mL bottle volume minus 60 mL of water leaves 40 mL headspace volume which is exchanged in < 1 min(?). Is this sufficient for establishing equilibrium given the applied flow rates? Were the tubes submerged?

*See comment on L95f and L133.*

L193: Was this the observed temperature range during sampling? Then 25°C (L197) may not be enough to prevent condensation.

*This was the temperature in the laboratory during storage. During the measurements, great care was taken to ensure that the temperature in the lab was higher than the temperature we measured during filling.*

*We will add this information and also discuss the implications for a wider use of the method.*

L218: Why did you test only the effect of one-day storage when you intended to store natural samples for up to seven days? Did you refill them with L22 before you "then proceeded" (L219) with H22? Why? Did you also assess the memory effect on samples stored in re-used bags for seven days after the previous samples had also been stored for that period? From your experience, what kind of preparation would be necessary in that case to still obtain meaningful isotope measurements from unknown samples stored in re-used bags?

*For our applications, the one-day period is the most interesting because we usually spend a day in the field taking measurements and then have time to analyze the next day.*

*Yes, we measured L22 after one day of storage and then refilled and measured again to make sure there was no effect on the same standard after one day of storage.*

*We did not perform a test with standards where we tested the memory effect after very long storage times (< 7d), as these long storage times were beyond the scope of the current experiment (but this could of course be explored in the future). However, we used the same bags for the field measurements in October and February (the field campaigns where we compared bag to in situ measurements) and were able to obtain good results after rinsing with dry air 10 times. In the natural abundance range, we therefore assume that this treatment works reliably for sampling.*

*We will discuss this in more detail in the revised manuscript.*

L220: What do you mean by "usual steps"? Did you refill with H22 and measure/empty immediately? How are the obtained findings transferable to a setting where, e.g., L22 was the first sample collected with a new bag and H22 was the sample collected with the (now reused) bag – also stored for 1 day, or 3 days, or 7 days? I am afraid, this is the weak point of the entire reusability test. By emptying the bags overnight (L223), you avoided exactly the effects that samples in reused bags may be subjected to. The point of reusing bags for unknown samples collected remotely should be to NOT have to refill/empty them repeatedly with the sample of interest and then measure them immediately. Can you propose a preparation routine for to-be-reused bags that ensures the isotopic composition of any unknown sample to be reproducible with sufficient accuracy after typical storage times? If not, I am afraid, the combined storage and memory test is not very

exhaustive. (Later, you suggest rinsing 10 times with dry air but you do not present data proving the usefulness of that procedure.)

*Thank you for your comment. It is true that we first had L22 in a new bag for one day, and then H22 was filled, measured, and emptied directly. We agree that rinsing 10 times with dry air is not completely transferable, but our recommendation was based more on our results from the field experiment in February, where we followed exactly this procedure. However, we see that this is not fully explained in the current version of the manuscript and the difference in the isotopic signal of the samples is not as strong as for the two standards (see section 3.3 combined storage and memory test). For this reason, we will 1) explain our field experiments in more detail and 2) perform an additional experiment following our field protocol in which we will store one standard in new bags for one day, rinse the bags with dry air, and then fill them with the opposite standard. We will then measure these samples one (and 3) days later and additionally present and discuss these results in the updated manuscript. (Unfortunately, we were not able to perform and present this test immediately, as we do not have all the necessary materials in stock). This will give more insights into the reusability of our method under different experimental settings (e.g. natural abundance vs. labelling approaches).*

L229: Please state here already, if you used new or reused bags for this part of the study.

*We will explain that we used new bags in October and reused bags in February. We will also adapt the graphics for a better understanding.*

L234f: This sentence sounds odd. Either insert "samples" after "45 cm" or delete "for" and change "taken" to "sampled"

*Will be changed.*

L239: Equilibrium is not indicated by stable values. Steady-state conditions are indicated by stable values. One way to test for equilibrium conditions is to vary the flow rate around the target value and see if this has an effect on readings of vapor mixing ratio and isotope signatures. Was this done?

*We tested the standard bottles used starting with the minimum flow the picarro needs to operate (around 35 ml / min) and increased the flow up to 300 ml / min. Until around 100 ml / min (75 ml / min + picarro flow), it resulted in accurate results. With 100 ml / min + picarro flow the water concentration started to decrease slightly (with still acceptable results). A higher flow rate of 150, 200 and 300 ml/min + picarro flow then resulted a depletion of heavy $^{18}O$ and $^2H$ isotopes relative to the standard.*

L241: What was the time per in situ measurement (as compared to 15 min of bag filling)?

*During this part of the experiment, we did at least 15 minutes of in situ measurements.*

L242: The logger only records the readings from an attached sensor. What sensor was connected to the logger to obtain temperature measurements?

*The sensor information will be added.*

L243: Please, also state here the durations of the individual steps. Most importantly, how long were samples stored in the reused(?) bags prior to measurements? How does this compare to the combined storage and memory test? And how is this transferable to a setting with no field-access to an analyzer? (I understood that bag measurements were conducted in the field shortly after filling.)

*In October, we first measured in-situ, then filled and measured the bags in the field, and remeasured them 1 day later in the lab. In February, in situ measurements were made in the field before filling and bags were measured in the laboratory the next day. We see that this part is not well explained here. We will rewrite this section (as well as the results for this experiment).*

L245: This statement is a repetition of L231f. Consider deleting.

*Will be changed.*

L282 (and elsewhere): For consistency, delete quotation marks for the names of the standards (here: L22 and M22).

*Will be changed.*

L290ff: This seems to be a repetition of the previous statement. Rephrase or delete

*Will be changed.*

L302f: "increased deviation" translates to high inaccuracy, not "imprecision". Accuracy describes the deviation from the target value and is not synonymous with precision, which describes the scatter of repeated or replicate measurements around a common mean.

*We have changed this paragraph in response to your comment. It now reads:*

*"The second storage test using L22, showed a lower accuracy (which was - 19.9 ‰ $\delta^{18}O$ and - 148.1 ‰ $\delta^2H$) being − 0.1 ± 1.1 ‰ for $\delta^{18}O$ and 2.8 ± 4.9 ‰ for $\delta^2H$. No trend could be observed, similar to the previous experiment. The lower accuracy was mostly caused by the increased inaccuracy after three days, as all gas bags showed a significant enrichment (8.9 ± 2 ‰ on average). The z scores show the same result with accurate values for $\delta^2H$ (except after 3 days) and a lower precision with questionable values for $\delta^{18}O$. The average z-score was 0.3 ± 2.7 for $\delta^{18}O$ and 1.4 ± 2.5 for $\delta^2H$ (see Table 3 for detailed values)."*

L303: insert "samples from" after "as".

*Will be changed.*

L305: please elaborate on the "error during measurement". What went wrong and how can users of your method avoid this error?

*We will discuss the error in the revised manuscript and recommend ways to avoid errors.*

L312: I don't think it is fair to compare the accuracy of two methods that used totally different storage times (1-7 days vs. 30 days).

*This is correct and we will balance the comparison in the revised version.*

L321: Given that Magh et al. (2022) used off-the-shelf components, I tend to say that their method is not more difficult to handle than yours. Further, the "static properties of the glass vials" (L322f) make overfilling impossible during sampling (as compared to a mandatory maximum of 90% in the case of the gas sampling bags used in this study) and allow for simply letting dry air flow in during measurement with no need of extra pumping. Apart from potential breaking, glass vials may also be more robust relative to the thin plastic and aluminum layers of sampling bags in many typical field settings (you report damaged bags yourself (L407)).

*It is true that there are both advantages and disadvantages in handling, preparation and analysis compared to the system proposed by Magh et al. (2022), which we will discuss in more balanced way. See also comments and replies above.*

L329ff: Personally, I find it alarming when the standard closest to ambient air delivers the best results as it points to an unaccounted-for influence of ambient air. The question must be how you can ensure that your method delivers meaningful results regardless of the isotopic composition of standards or samples. And how does this impact the measurement of unknown field samples when collected using newly prepared, equally pre-treated bags?

*This is of course true, but as we already wrote in L335-337: "The overall higher scatter (particularly for $\delta^{18}O$) visible in the experiment using standard L22, which has a different isotopic signature than the ambient air, led to initial concern over potential exchange with ambient air. However, we do not think that is likely as the visible scatter already appeared within one day of storage, was not directed towards isotopic signatures of ambient air and did not increase over time."*

L337: No. Flushing with dry air in the case of Herbstritt et al. (2023) did not cause the scattering. Rather, it was unsuitable to remove the scattering caused by previously collected, diverse samples as efficiently as flushing with moist air did.

*This statement will be adapted in the revised manuscript with the separation of results and discussion.*

L353: The connection between storage time and memory effect has already been shown in the Herbstritt study.

*This statement will be adapted in the revised manuscript with the separation of results and discussion.*

L356f: Insert "target" or equivalent before "standard deviation" (2x).

*Will be changed.*

L363: I don't know which part of the Herbstritt study you are referring to but as I understand they used ambient, non-saturated air of arbitrary isotopic composition to pre-condition their bags.

*That's correct. It now reads: "In their study, the bags were additionally pre-flushed with ambient air of a known isotopic signature."*

L377f: Clearly, the magnitude is a function of the isotopic spread between the standards used here. The exponential decrease – expressed in the standard deviation of an entire batch of to-be-reused bags – was also shown before (Herbstritt et al., 2022, Fig. 5b).

*We will include/clarify this in more detail in the revised discussion.*

L379f (and elsewhere): I think it is not necessary to repeat the isotopic composition of the standards so often. Ideally, the outcome of your method should be independent of these values anyway.

*Will be changed.*

L382: Why did you stop at H7? It would also be important to confirm that the readings stay in that range.

*The measurements during this experiment took a long time, which meant that we were only able to carry out 7 repetitions within two days. As H5 and H6 were already close to the accurate range, we decided not to carry out any further measurements.*

L397ff: You advise to reuse bags but you did not show how the isotopic signature of unknown samples can be obtained in the foreseen application, i.e. remote sampling followed by lab-based analysis on a different day. In the storage and memory test you repeatedly flushed the reused bags with standard vapor until the readings were acceptable (after irrelevantly short "storage" times). The proposed procedure (filling and emptying at least seven times (L400) and promptly measuring) is certainly not desirable (or feasible) when collecting unknown samples in remote locations. What would be the achievable sampling frequency in that case? And would that still be an advantage compared to direct in situ measurements performed with an analyzer that has been brought to the field?

*Thank you for your comment. We understand that with the explanations and results presented in this form, an unrestricted recommendation for reuse cannot be made. By splitting the October/February measurements with the additional explanation that rinsed and reused bags were used in February, we can currently only recommend this method for measurements in a narrow natural abundance range (and following strict guidelines, see above). We will also perform an additional experiment (see comment above) to be able to make a statement about samples with larger differences in isotopic signature.*

*But to answer your questions for possible future experiments: Filling the sample bags ten times with the target sample in the field and then emptying them would make the system more complex, as one pump would be needed for filling and one for emptying. However, if a system were built for each sample bag that automatically fills (~15 minutes) and empties (~1 minute) the bags and collects the samples at the same time (you would need as many pumping systems as you have samples), such sampling could be done in about 3 hours with a theoretically unlimited number of samples.*

*We will include detailed suggested sampling protocols in the revised version.*

L400: With what and for how long should re-used bags be filled? I am sure this has in impact on feasible sample storage time. Can you also comment on a quantitative relationship between the ranges in isotopic compositions of previous samples and the necessary number of pre-sampling filling cycles?

*The bags were rinsed with dry air. This statement will be adapted in the revised manuscript with the separation of results and discussion.*

L404: Did you compare in situ measurements and bag measurement only during two or during all 18 campaigns? If two, then how were conditions different, especially regarding elapsed time between sampling and measurement and relative to the sample storage time tested in the combined experiment? Please specify in the method section.

*This statement will be clarified in the revised manuscript with the separation of results and discussion. In addition, we will add a more detailed explanation of the experiment in the methods section for a better understanding.*

- *Yes, only two of the 18 campaigns compared in situ and bag measurements.*
- *In the first campaign, we first measured in situ and then the bags immediately after filling (resulting in a direct/bag measurement in ~30 minutes) as well as one day later in the lab.*
- *In the second campaign, we measured in situ and filled the bags. The bags were then measured in the lab within 24 hours after filling.*

L407: To make life easier for potential users of your method, please specify "filling errors". In addition, how did you identify condensation? Where did you see it?

*We will add a section to the discussion that explains/discusses filling errors and how to handle them. Regarding condensation, we once measured a bag at a temperature that was too low (the AC flow was directed toward the bag), resulting in a small condensation peak during the bag measurement. Since we could not be sure that there was no effect on the rest of the sample, we discarded this bag. Condensation during bags filling should be avoided by flushing the soil probes in the field with dry air prior to the measurement.*

L409: This is important and should appear in the method section already: What did you use for rinsing the bags and where was this step performed? Standard-derived vapor in the lab or the to-be-collected, unknown sample vapor in the field? If the latter, what was the required per-sample time required for this step? 10 x 15 min = 150 min?

*We used dry air to rinse the bags. We will explain our handling in more detail in the Methods section and later in the Results/discussion section.*

L432: On what kind of analyzers do co-extracted organic compounds interfere with water stable isotope measurements?

*Laser based cavity ring down spectrometer like the CRDS we used (Picarro 2310-i). We will clarify this statement.*

L444: After what?

*Will be deleted.*

L446: Please specify "wide"

*We will add the "wide range" in a bracket.*

L447: The period needs to be specified.

*Will be changed. It now reads: "The isotopic signature of precipitation is represented by the local meteoric water line (LMWL), shown here for the period of September 2021 to September 2023."*

L455: For additional plausibility, can you compare the nature of the scatter, e.g., by comparing the linearity (R²) of the dataset, with that of precipitation data and other datasets of soil water isotopes? Is there a difference in linearity between the two campaigns with field-access to the analyzer and the other 16 without (if that was the difference)? How were standards produced and treated in these two different cases? How many validation standards were co-measured and what was their precision and accuracy?

*We will change the graphic to better show the different campaigns and add a more detailed comparison/explanation of the different depth and seasonal development. Three laboratory standards were bagged and treated in the same manner as the samples.*

L458: transpiration rather does not cause enrichment. Evaporation does. Please change "evapotranspiration" to "evaporative"

*Will be changed.*

L462f: Where do I find the seasonal variability you are referring to?

*Will be changed. It now reads: "Overall, our findings from the field trial suggest a good agreement with GPM probe and bag-based soil water isotope measurements with the LMWL and are plausible in terms of seasonal variability (see Fig. 6c; e.g. compare offsets between cryogenically extracted bulk soil water isotope measurements and LMWL; e.g. Zhao and Wang, 2021)."*

L465f: This seems to be a bit off. Usually, the lower boundary of the plow layer is around 20 cm, not 45 cm. Was it different in your case? Can you also comment on the large range of isotope values observed for 150 cm depth (yellow symbols in Fig. 6)? I would expect to see a less pronounced variation at that depth.

*It's correct that the lower boundary of the plow layer is typically located around 20 cm but it depends on the soil conditions during plowing (high soil water contents can lead deeper plowing). We actually expected the lower plow boundary to be 20 cm and consequently the deeper probes to be unaffected by tillage. Hence, the probes at 45 cm and 150 cm were not recovered and reinstalled before and after tillage. In comparison, we routinely remove/reinstall the soil probes in the upper layers (5cm and 15cm) during/after tillage. After discovering the very low vapor concentrations in the probes in 45 cm depths, we suspected damage to the probes due to the tillage. Personal communications with our field manager revealed, that the tillage was indeed deeper than 20 cm and likely resulted in a compaction of the soil down to the 45 cm probes. We have repeatedly tried to measure these probes and could measure some of them in a vapor concentration matching the vapor saturation at the given temperature. Those measurements were deemed likely to be valid and were included in the manuscript.*

*We will clarify this statement in the revised manuscript and add a more detailed discussion on the implications of soil manipulation for long-term use of the in situ systems.*

L468: Why does soil compaction flaw the measurements? In situ measurements have been conducted successfully in boreholes of (I would say: rather compact) trees by one of the co-authors. So why wouldn't they work in compacted soil? And why would that be an issue at 45 cm but not at 150 cm depth?

*See comment above. (The compacted soil is not the problem in itself only the fact that probes in 45 cm were installed before tillage i.e. were in the soil when the compaction occurred which is the*

*typical handling of sensors in many agricultural studies, e.g. only de-install sensors that are above the manipulation depth)*

L475: I think, "appropriate" is inappropriate here. You did not test the effect on samples stored in reused bags for more than 1 hour. (Or you forgot to mention that.) Consequently, I do not see how reliable measurements of unknown samples stored for typical time periods in reused bags can be performed based on the findings of this study.

*This statement will be adapted in the revised manuscript with the separation of results / discussion and considering the field experiment and the additional experiment.*

L476: rinsing with dry air does not match the procedure described in the combined memory and storage experiment. Please explain (before the conclusion), why rinsing with dry air – previously suspected to increase scatter – does (or should do) the same trick that flushing with moist air does.

*This statement will be adapted in the revised manuscript with the separation of results and discussion. See also comments and replies above for specifics.*

L485: are these numbers based on two or on 18 campaigns?

*These numbers are based on the two campaigns of in situ and bag measurements. We will adjust this statement in the revised manuscript with the separation of results and discussion.*

L490: Not "can" but "will"

*Will be changed.*

S1: AMT is a European Journal. I suggest using the metric system and SI units.

*Will be changed to SI units.*

S2 & S3: What depths are you referring to? Weren't these measurements performed on standard vapor sampled in the lab?

*"Depth" will be deleted. It now reads: "Differences during the storage experiment for M22 and L22 for each storage duration…"*

---

## Author Response (AR1)

**Reply to reviewer 1 on manuscript amt-2024-43:**

**Review on manuscript amt-2024-43**

Title: An easy-to-use water vapor sampling approach for stable isotope analysis using affordable membrane valve multi-foil bags

Author(s): Adrian Dahlmann et al.

**General comments:**

In the manuscript by Dahlmann et al. the authors present an alternative approach (in comparison to Herbstritt et al., 2023, Magh et al., 2022 and Havranek et al., 2021) of obtaining water vapor samples for analyzing the isotopic composition of soil water obtained with gas-permeable membranes (GPM) and storing them in multi-layer foil bags. They performed different experiments to test maximum storage time, potential memory effects and reusability as well as field applicability. The authors conclude that their approach is a simple, cost-effective, and versatile approach.

The paper is nicely written and well structured.

However, I have two main concerns:

As approach and concept in this manuscript are very similar to the paper by Herbstritt et al. (2023) - only a different type of GPM and a different (commercially available) bag type were used - the results should be compared point by point and discussed accordingly. Here some revisions and additional considerations are needed.

Moreover, the cited study by Jiménez-Rodríguez is only available as preprint in HESSD. It was under review in 2019 but not accepted due to substantial issues. No revision was provided by the authors afterwards, thus no accepted peer-reviewed version exists. This pre-print can therefore not be cited in your manuscript.

*Thank you very much for the positive feedback on our work. We now revise the manuscript in detail to answer all your questions and those of the second reviewer. As part of this revision, we removed all statements/references to the preprint by Jiménez-Rodríguez. In addition, as recommended by reviewer 2, we separated the discussion/results and include a detailed comparison with Herbstritt et al. (2023) and other comparable methods, as well as a detailed recommendation about 1) the use of our method, 2) possible errors and how to avoid them, and 3) what should be tested for future use (in detail in the supplements "Handling Recommendations").*

**Specific Comments:**

1. L28: what you call 'precision' is the deviation from the true value, isn't it?

   *Yes, we changed it to "accuracy" for a better understanding and changed the wording according to the ISO 5725 definition throughout the manuscript.*

2. L30: (and throughout the manuscript) 'water stable isotopes' not 'stable water isotopes'

*Done.*

3. L34: 'suitable for many applications' was already mentioned in L. 31. Please rephrase

   *This sentence was deleted in the course of the revision.*

4. L61: 'which can cost anywhere from' sounds a bit sloppy

   *It now reads: "To do so, primarily glass bottles or gas sampling bags with various fittings are used, which cost from ~1-200 euros per container."*

5. L67: '…provide new insights in research' please be a bit more specific or delete.

   *Done. It now reads: "These simplified and more affordable systems could therefore increase the number of studies on stable water isotopes and provide new insights in research by increasing the number of possible experimental sites and samples."*

6. L98: wrong! ‰ is not "million"

   *Changed.*

7. L112: P/N of bags? Fitting of bags? Volume of bags?

   *We could not find a part number but we now added a link to the bags and the product name on the website (Multi Foil Bags with Stainless Steel Fitting, https://www.smelltest.eu/en/product/multi-foil-bags-with-stainless-steel-fitting/). All the information we received is now included here or in Table S1 in the supplement.*

8. L115: use the manufacturer as reference for material properties such as 'Water Vapor Transmission Rate'

   *See comment above.*

9. L125: Fig. 1: Where is the bag? On the right side in Fig. 1b?

   *Yes, the bag is on the right side of Fig. 1b. In Fig. 1a, the connector is disconnected from the bag. We now explain this in the description of the Figure.*

10. L130: how much standard water was added to the 100 mL bottle? And what was the size of the GPM inside the bottle? This is crucial for the isotopic equilibrium

    *Thanks for your comment and question. The standard measurement is now explained (see section 2.1 Study area and basics of water stable isotope measurements).*

11. L275: Jiménez-Rodríguez can't be cited (see general comments)

    *All statements/references to Jiménez-Rodríguez's paper have been removed.*

12. L289: Fig. 3: would be nice if you could add "L22" and "M22" directly to the figure, there is enough white space. The Legend is also a bit small.

*Changed.*

13. L305: if you assume an error during the measurement, I'm wondering why do you present these data and didn't repeat the measurement?

   *We did not repeat the measurement mainly because the results after 7 days looked promising and supported our reasoning against a potential storage effect. Following recommendations of reviewer 2, we will split the results and discussion and include all recommendations to avoid problems in the supplement ("Handling Recommendations").*

14. L325 and L332: Jiménez-Rodríguez can't be cited (see general comments)

   *All statements/references to Jiménez-Rodríguez's paper have been removed from the revised manuscript.*

15. L324 to L342: please rephrase and take into account the findings of Herbstritt et al. (2023)

   *We now discuss our results and how they differ from Herbstritt et al. (2023).*

16. L354: Fig. 4: see comments to Fig. 3

   *Changed.*

17. L378: "...has not been described in the literature before" please rephrase or delete. I would say it is comparable to Herbstritt et al. (2023), Fig. B1.

   *Changed.*

18. L378-400: Please discuss and compare the results of Herbstritt et al. (2023) on memory effects and conditioning here.

   *We now discuss our results and how they differ from Herbstritt et al. (2023).*

19. L504 ff: Please check the reference list thoroughly, e.g. Millar et al., (2018) or Orlowski et al., (2016b) are in the reference list but not cited in the text.

   *Thank you very much. All references have been checked.*

**Technical correction:**

1. L11: 'water stable' not 'water-stable'

   *Done.*

2. L16: 'easy-to-perform' instead of 'easy to perform'

   *Done.*

3. L30: I suggest "…using GPM combined with…" instead of "…using GPM and…"

   *Done.*

4. L34: 'suitable' instead of 'suited'

   *Done.*

5. L83: please use different bullet points for the field experiments e.g. i) ii) iii) or (a) (b) (c)

   *Done.*

6. L98: wrong wording: detailed in

   *Done.*

7. L108: Majoube is from 1971

   *Done.*

8. L215: 'procedure similar to' instead of 'similar procedure to'

   *Done.*

9. L247: please delete line break

   *Done.*

10. L284: I suggest 'On average' instead of 'In average'

    *Done.*

11. L396: 'especially in' instead of 'in especially'

    *Done.*

12. L447: please add dates

    *Done.*

**R2: General comments**

The authors present a new method of collecting discrete vapor samples for water vapor stable isotope analysis using inflatable multi-foil bags. The presented method contributes to a new, currently evolving field of stable isotope analysis still lacking an agreed-upon standard procedure suitable and attractive for many users interested in performing in situ isotope assays without field-access to an analyzer. Therefore, any reported experience in this regard is highly welcome and I recommend publication after proper revision.

The manuscript describes the use of bags, which differ only in valves (which do not seem to have an effect) from the ones used in a previous study (Herbstritt et al., 2023, doi: 10.5194/hess-27-3701-2023). I therefore suggest a more thorough discussion emphasizing how this work expands the findings of the previous study. Moreover, I don't understand how the proposed treatment of previously used bags would help to get meaningful results if reused for unknown samples. I have a feeling that the tested treatment to remove memory effects does not account for the potential conditions faced by researchers interested in using the proposed method regarding, e.g., feasible or necessary storage time and range of previously observed isotope values.

Formally, the authors decided to combine results and discussion. Unfortunately, this often leads to a limited description of the results. I believe the manuscript would benefit from a better distinction between description and interpretation of the presented findings. Also, some additional technical details (flow measurement devices, new or reused bags for the field test, rinsing atmosphere prior to reuse, etc.) should be added to the method section. Finally, a detailed SOP listing suggested settings and potential pitfalls may be helpful for future users of the proposed method.

*Thank you very much for your detailed and constructive comments and for recommending publication. We have thoroughly revised the manuscript to explain in detail all questions raised in the comments or to clarify misunderstandings. In particular, we have highlighted the differences between our work compared to previous studies more clearly:*

*In situ measurements of water stable isotopes are usually performed with two different systems: the recently commercially available WIP system (as used in the study by Herbstritt et al., 2023) and originally developed by Volkmann et al., (2014; for soils) and (2016, for xylem of trees), and home-built systems with GPMs (following the original developments of Rothfuss et al., 2013 and as used in Kübert et al., 2020 or Kühnhammer et al., 2021). The main difference between these systems is that the WIP system dilutes the sample flow by reducing the water vapor concentration in the probe, hence enabling measurements with relatively constant water vapor concentrations. Home-built systems with GPMs usually measure the saturated airflow without dilution in the GPM. One of the main differences is that the water vapor concentration of a sample from the WIP system is usually lower than that of the self-built systems due to the dilution. This has the advantage of reducing the risk of condensation, but also leads to a lower water concentration and thus a reduction in sample volume. We believe that aside from the material used for the storage container (different types of bags, glass vials etc.), the in situ method itself is also an important part that can influence the method development of a new storage method and find it relevant to compare our approach using a self-constructed in situ system with the WIP system used in Herbstritt et al. 2023. We*

*have clarified more clearly on this and all other important differences between our work and that of Herbstritt et al, (2023) during the revision of the manuscript.*

*Moreover, we agree that rinsing 10 times with dry air is not completely transferable, but our recommendation was based more on our results from the field experiment in February, where we followed exactly this principle (rinse used bags from October 10 times). However, we see that this was not fully explained in the first version of the manuscript and that the difference in the isotopic signal of the samples was also not nearly as strong as for the two standards (see old section 3.4 Field test), which limits the recommendation. For this reason, we have 1) changed the explanation of our field experiments to avoid misunderstandings and 2) performed an additional experiment following our field protocol where we will store one standard in new bags for one day, rinse the bags with dry air, and then fill them with the opposite standard. We then measured these samples one (and 3) days later and additionally discussed and presented these results in the updated manuscript (Discussion/Supplement). This gives more insight into the reusability of our method in different experimental settings (e.g. natural abundance vs. labelling approaches).*

*As recommended by the reviewer, we have split the Results and Discussion sections to make the results more understandable and to discuss them in more detail. In the Discussion, we include all the information requested by the reviewers and compare our results with those of other studies (especially Herbstitt et al., 2023, but also Magh et al., 2022 and Havraneck et al., 2020). Throughout the whole manuscript, we have added information to make the differences more explicit (e.g. home-made GPM vs. WIP system or glass bottles vs. bags or lab vs. field experiments or different storage times) and to point out important advantages and disadvantages in a more comparative way. In addition to these comparisons, we have added a section on how to use our system and how to avoid potential problems ("Handling Recommendations").*

I provide a list of specific comments below.

**Specific comments**

L10: "water stable", not "water-stable"

*Changed.*

L16: "easy-to-perform, in situ", not "easy to perform, in-situ" (also elsewhere in the MS: "in situ" without hyphen)

*Changed.*

L22: insert "spectrometer" or equivalent after "laser"

*Changed.*

L25: "can lead" seems too weak, as there will always be influence of previous samples. I suggest "does lead" or "will lead"

*This sentence has been deleted.*

L26: Consider rephrasing to: "…showed that the memory effect increases with duration of storage."

*Changed.*

L28: You state the precision, which describes the scattering of repeated or replicate measurements. What is the accuracy, i.e. the deviation from the target value?

*We changed the wording according to the ISO 5725 definition throughout the manuscript.*

L30 (and elsewhere): "Water stable", not "stable water"

*Changed.*

L38f: I do not see why hydrology and meteorology would focus on the biosphere. Consider rephrasing.

*Changed.*

L59 (and elsewhere): Do not cite preprints like Jimenez-Rodriguez et al. (2019). It is against AMT guidelines and it devalues the work of reviewers. Especially, do not call such work "successful" (L70) when in fact it has not successfully passed a peer-review process.

*All statements/references to Jiménez-Rodríguez's paper will be removed from the revised manuscript.*

L61: "less than 50 euros" is quite vague. Can you be more specific?

*It now reads: "To do so, primarily glass bottles or gas sampling bags with various fittings are used, which cost from ~1-200 euros per container."*

L63: in what aspect is the lab environment stable? – Temperature?

*Yes. It now reads: "temperature-stable laboratory environment".*

L63: What do you mean by "configuration"?

*This can be misleading due to our wording. By "time-consuming configuration" we meant the time-consuming post-procession step (calculation) to obtain the true isotopic value. Using the glass bottle method requires the filling of missing sample volume with dry air to avoid suction inside the glass bottle. Naturally, this involves an additional correction step, due to the vapor concentration significantly declining over the measurement period (see section 2.1 VSVS lab test in Magh et al., 2022).*

*In our opinion and after testing the glass bottle method ourselves in collaboration with part of the co-author team of the Magh et al., paper, we concluded, that a storage method based on bags rather than glass vials is easier to handle. With the gas bag method, isotopic signatures of the gas inside the bag are directly recorded with a stable water vapor concentration.*

*Of course, this does not necessarily imply that one method is superior to the other, hence we rewrote this section.*

L72: Again, be more specific about pricing. This helps other researchers considering using your method. In addition, the numbers never appeared in the manuscript again, i.e. they were not discussed. Nonetheless, you refer to them prominently in the manuscript's title. How do they compare to the per-sample costs of the containers used by Magh et al., (2022, doi: 10.5194/hess-26-3573-2022) and Herbstritt et al. (2023)?

*We have added the approximate actual cost per bag in parentheses (costs have increased since we purchased our materials).*

L95f: Please be more specific. How was vapor from standards produced? Was it in equilibrium with the liquid phase (resulting in temperature-dependent isotope fractionation) or flash-evaporated (with no fractionation)?

*Thanks for your comment and question. The standard measurement is now explained (see section 2.1 Study area and basics of water stable isotope measurements).*

L98: per mil, not parts per million (I wonder how this went unnoticed by five co-authors…)

*Changed.*

L108: The Majoube paper is from 1971, not 1961.

*Changed.*

L112f: Would it be possible to state a part number for these bags as well? I am unable to find this product in a web query. In addition, how does a membrane-based valve work? Does the sample have to pass through a membrane?

*We could not find a part number but we now added a link to the bags and the product name on the website (Multi Foil Bags with Stainless Steel Fitting, https://www.smelltest.eu/en/product/multi-foil-bags-with-stainless-steel-fitting/). All the information we received is now included here (section 2.2.1) or in Table S1 in the supplement.*

*It now reads: "The stainless steel 2-in-1 fitting combined the valve and septum, with the septum acting as a seal, allowing air to flow around it when the valve was open, and sealing when the valve was closed."*

L115: This number seems to be huge! Assuming that the sample bags (front and back) have an area of roughly one tenth of a square meter, more than half of a sample (which comprises ~17μL or 17mg of water per 1 L air at room temperature when saturated) would be exchanged per day. Can this be true? Please, also state the conditions (temperature, relative humidity, vapor pressure gradient), under which the water vapor transmission rate was determined (without citing a preprint). Otherwise, this number is meaningless. Or disturbingly high.

*Thank you for your comment. In fact, this number is not correct. We calculated the WVTR again and the correct value is 0.00465 gr/m2/24h. Here is the manufacturer's information and the calculation in metric system:*

*Water vapor transmission rate (FED 101): < 0.0003 gr / 100 in² / 24 hrs*

- *100 in² = 0,064516 m² → 1550 in² = 1 m²*

- $< 0.0003 \ gr \ / \ 100 \ in^2 \ / \ 24 \ hrs \ * \ 15.5 = \ < 0.00465 \ gr \ / \ m^2 \ / \ 24h$

*With a bag area of ~ 640 cm² it would be:*

- $< 0.00465 \ gr \ / \ m^2 \ / \ 24h \ * \ 0.064 = 0.0002976 \ gr \ / \ bag \ area \ / \ 24h$ *or*
- $< 0.2976 \ mg \ / \ 24h$ *for a bag.*

*With 15.3 mg of water sample in 0.9 L of air at room air temperature at saturation, this would be ~ 2% per day or ~ 14 % per week, but (as you already mentioned in your second question about the conditions) this is an extreme value tested with the "FED-STD-101 – Test Procedure for Packaging Materials" at high water concentrations (90%) on one side and low water concentrations (desiccant) on the other side at ~ 38°C (http://www.woodencrates.org/standards/FED-STD-101.pdf).*

*All statements have been corrected.*

L121: Did you test a version without electrical isolation tape that did not work? I am wondering if the tape really makes a difference regarding proper sealing.

*It is true that the electrical tape per se is not important for proper sealing. Initially, we tested the bags without tape, but the adhesive in combination with the PTFE tubing can break under tension, which (of course) leads to leakage. Therefore, we used the electrical tape to stabilize the connector (you could probably use any tape, but we had the electrical tape in abundant stock). We have explained this in the revised manuscript.*

L127: What was the length of the GPM?

*The length of the GPM is not as important here (< 5 cm), as the dry air passes the standard water, and it is more of a safety mechanism to prevent liquid water from entering the tube/analyzer. In the field experiment, we used approx. 12 cm GPM (comparable to soil GPM in e.g. Kühnhammer et al., 2021).*

*The measurements are now explained (see section 2.1 Study area and basics of water stable isotope measurements and 2.4 Experimental design).*

L133: How was the flow rate measured? And what would have been the maximum possible flow ensuring equilibrium given the GPM length you selected?

*Thanks for your comment and question. All information is now included in section 2.1.*

*It now reads: "A gas cylinder was used to induce dry gas at a low flow rate of 50 - 80 ml per minute (257-6409, RS Components GmbH, Germany). We ensured that the isotopic signature of the vapor would be at equilibrium with liquid water at this flow rate. We tested flows from the minimum required for Picarro operation (approximately 35 ml/min) to 300 ml/min and found accurate results to 100 ml/min."*

L135: Under non-EQ conditions, the vapor isotopic composition would also depend on water isotopic composition and surrounding temperature. But not exclusively.

*All information is now included in section 2.1.*

L140: By "outgoing", do you mean the flow going out of the sample vessel or the flow going out of the open outlet?

*We are talking about the "open split". We changed it for a better understanding:*

*"Since the laser spectrometer only has a flow rate of approx. 35 to 40 ml per minute, an open split was added to ensure a constant flow and to avoid pressure differences. The open split was continuously measured to ensure that no ambient air could flow back."*

L163: How dry was the air after passage through the desiccant? Was this value tested and constant over the course of the experiment?

*It now reads: „The dry air supply box was tested prior to our experiments by measuring the outlet concentration of the dry box over the course of one day. However, the use of such a system should always be tested for the specific application, as a very high flow rate combined with very humid air could greatly affect the duration of possible use. During the experiments, we periodically tested the water concentration before and after the field campaigns and could not detect any increase after one day in the field. The water concentration of the dry air produced was approximately 200 ppm."*

L166f: What would happen, if the bags were filled to more than 90% capacity? And why isn't a lower filling capacity stated in the first place? How about filling only to the minimum volume necessary to reach a plateau on the analyzer during analysis? Did you play with that variable as well? How would that impact feasible sample throughput? How would the reduced sample volume affect its vulnerability, e.g., regarding memory effects?

*Thank you for these interesting questions. We now mention it in the beginning of the method section and give detailed information about bag handling in the supplement („Handling Recommendations").*

*As recommended by the manufacturer, care was taken when filling the bags to ensure that the maximum volume did not exceed 90% of capacity, which could cause material damage.*

*Personally, we do not recommend a lower filling quantity, as this could change the volume to area ratio and increase the effect of the water vapor transmission rate. In turn, this could potentially increase storage and memory effects.*

*A reduced sample volume could potentially have a positive effect on sample throughput in the field, as the filling time would be significantly reduced. However, a higher sample throughput could also be achieved by simply using multiple dry air pumps, i.e. filling the bags simultaneously in the field, without having to reduce the sample volume.*

L173ff: This statement is a repetition of L141f. Consider deleting.

*Changed.*

L180: 100 mL bottle volume minus 60 mL of water leaves 40 mL headspace volume which is exchanged in < 1 min(?). Is this sufficient for establishing equilibrium given the applied flow rates? Were the tubes submerged?

*See comment on L95f / L133.*

L193: Was this the observed temperature range during sampling? Then 25°C (L197) may not be enough to prevent condensation.

*This was the temperature in the laboratory during storage. During the measurements, great care was taken to ensure that the temperature in the lab was higher than the temperature we measured during filling.*

*Detailed temperature information is now explained in section 2.1 for all experiments.*

L218: Why did you test only the effect of one-day storage when you intended to store natural samples for up to seven days? Did you refill them with L22 before you "then proceeded" (L219) with H22? Why? Did you also assess the memory effect on samples stored in re-used bags for seven days after the previous samples had also been stored for that period? From your experience, what kind of preparation would be necessary in that case to still obtain meaningful isotope measurements from unknown samples stored in re-used bags?

*For our applications, the one-day period is the most interesting because we usually spend a day in the field taking measurements and then have time to analyze the next day.*

*We now explain the experiments in more detail and provide information in the discussion about the memory effect after a longer storage period (3 days) and larger isotope differences (additional experiment). In addition, all the information we obtained from working with the bags is contained in the manuscript and in detail in the "Handling recommendations" (supplement).*

L220: What do you mean by "usual steps"? Did you refill with H22 and measure/empty immediately? How are the obtained findings transferable to a setting where, e.g., L22 was the first sample collected with a new bag and H22 was the sample collected with the (now reused) bag – also stored for 1 day, or 3 days, or 7 days? I am afraid, this is the weak point of the entire reusability test. By emptying the bags overnight (L223), you avoided exactly the effects that samples in reused bags may be subjected to. The point of reusing bags for unknown samples collected remotely should be to NOT have to refill/empty them repeatedly with the sample of interest and then measure them immediately. Can you propose a preparation routine for to-be-reused bags that ensures the isotopic composition of any unknown sample to be reproducible with sufficient accuracy after typical storage times? If not, I am afraid, the combined storage and memory test is not very exhaustive. (Later, you suggest rinsing 10 times with dry air but you do not present data proving the usefulness of that procedure.)

*Thank you for your comment. It is true that we first had L22 in a new bag for one day, and then H22 was filled, measured, and emptied directly. We agree that rinsing 10 times with dry air is not completely transferable, but our recommendation was based more on our results from the field experiment in February, where we followed exactly this procedure. However, we see that this was not fully explained in the first version of the manuscript and the difference in the isotopic signal of the samples is not as strong as for the two standards (see experiment III). For this reason, we have separated our field experiment from the field data and explained it in more detail (2.4.3). In addition, we performed an experiment following our field protocol, in which we stored one standard in new bags for one day, rinsed the bags with dry air, and then filled them with the opposite standard. We then measured these samples one (and 3) days later (see discussion for more details).*

L229: Please state here already, if you used new or reused bags for this part of the study.

*We now explain that we used new bags in October and reused bags in February.*

L234f: This sentence sounds odd. Either insert "samples" after "45 cm" or delete "for" and change "taken" to "sampled"

*Changed.*

L239: Equilibrium is not indicated by stable values. Steady-state conditions are indicated by stable values. One way to test for equilibrium conditions is to vary the flow rate around the target value and see if this has an effect on readings of vapor mixing ratio and isotope signatures. Was this done?

*As this method of sampling soil water isotopes is well established and has been used by us and several other studies, we did not carry out an additional test. However, when the system was set up, we found no effect on the signal with different airflow rates below 100 ml per minute.*

L241: What was the time per in situ measurement (as compared to 15 min of bag filling)?

*During this part of the experiment, we did at least 15 minutes of in situ measurements.*

L242: The logger only records the readings from an attached sensor. What sensor was connected to the logger to obtain temperature measurements?

*We added the sensor information.*

L243: Please, also state here the durations of the individual steps. Most importantly, how long were samples stored in the reused(?) bags prior to measurements? How does this compare to the combined storage and memory test? And how is this transferable to a setting with no field-access to an analyzer? (I understood that bag measurements were conducted in the field shortly after filling.)

*These questions arose from our misleading description of the experiments and should now be clarified.*

L245: This statement is a repetition of L231f. Consider deleting.

*Changed (Rewritten at top).*

L282 (and elsewhere): For consistency, delete quotation marks for the names of the standards (here: L22 and M22).

*Changed.*

L290ff: This seems to be a repetition of the previous statement. Rephrase or delete

*Changed.*

L302f: "increased deviation" translates to high inaccuracy, not "imprecision". Accuracy describes the deviation from the target value and is not synonymous with precision, which describes the scatter of repeated or replicate measurements around a common mean.

*We have changed this paragraph in response to your comment. It now reads:*

*"The second storage test using L22, showed a lower accuracy due to lower precision for $\delta^2H$, being $-0.1 \pm 1.1$ ‰ for $\delta^{18}O$ and $2.8 \pm 4.9$ ‰. However, no time trend was observed. The decreased accuracy was mostly caused by the samples after three days, as all gas bags showed a significant enrichment ($8.9 \pm 2$ ‰ on average). The higher inaccuracy after three days of storage must be due to an error during the measurement, as accuracy improved again after 7 days. The z scores show accurate values for $\delta^2H$ (except after 3 days) and more questionable values for $\delta^{18}O$. The average z-score was $0.3 \pm 2.7$ for $\delta^{18}O$ and $1.4 \pm 2.5$ for $\delta^2H$."*

L303: insert "samples from" after "as".

*Changed.*

L305: please elaborate on the "error during measurement". What went wrong and how can users of your method avoid this error?

*We did not repeat the measurement mainly because the results after 7 days looked promising and supported our reasoning against a potential storage effect. We have separated the results from the discussion and included all recommendations to avoid problems in the discussion and in detail in the supplement ("Handling Recommendations").*

L312: I don't think it is fair to compare the accuracy of two methods that used totally different storage times (1-7 days vs. 30 days).

*We now discuss this statement in a more balanced way.*

L321: Given that Magh et al. (2022) used off-the-shelf components, I tend to say that their method is not more difficult to handle than yours. Further, the "static properties of the glass vials" (L322f) make overfilling impossible during sampling (as compared to a mandatory maximum of 90% in the case of the gas sampling bags used in this study) and allow for simply letting dry air flow in during measurement with no need of extra pumping. Apart from potential breaking, glass vials may also be more robust relative to the thin plastic and aluminum layers of sampling bags in many typical field settings (you report damaged bags yourself (L407)).

*It is true that there are both advantages and disadvantages in handling, preparation and analysis compared to the system proposed by Magh et al. (2022), which we now discuss in a more balanced way. See also comments and replies above.*

L329ff: Personally, I find it alarming when the standard closest to ambient air delivers the best results as it points to an unaccounted-for influence of ambient air. The question must be how you can ensure that your method delivers meaningful results regardless of the isotopic composition of standards or samples. And how does this impact the measurement of unknown field samples when collected using newly prepared, equally pre-treated bags?

*This is of course true, but as we already wrote in L335-337: "The overall higher scatter (particularly for $\delta^{18}O$), which has a different isotopic signature than the ambient air, led to initial concern over potential exchange with ambient air. However, we do not think that is*

*likely as the visible scatter already appeared within one day of storage, was not directed towards isotopic signatures of ambient air and did not increase over time."*

L337: No. Flushing with dry air in the case of Herbstritt et al. (2023) did not cause the scattering. Rather, it was unsuitable to remove the scattering caused by previously collected, diverse samples as efficiently as flushing with moist air did.

*This statement has been deleted with the separation of results and discussion.*

L353: The connection between storage time and memory effect has already been shown in the Herbstritt study.

*This statement has been deleted with the separation of results and discussion.*

L356f: Insert "target" or equivalent before "standard deviation" (2x).

*Changed.*

L363: I don't know which part of the Herbstritt study you are referring to but as I understand they used ambient, non-saturated air of arbitrary isotopic composition to pre-condition their bags.

*This statement has been deleted with the separation of results and discussion.*

L377f: Clearly, the magnitude is a function of the isotopic spread between the standards used here. The exponential decrease – expressed in the standard deviation of an entire batch of to-be-reused bags – was also shown before (Herbstritt et al., 2022, Fig. 5b).

*It now reads: „However, when the water source was changed to H, there was a clear memory effect of a magnitude up to -4.9 ± 1 ‰ δ18O in and -37 ± 6.4 in ‰ δ2H (Fig. 5 and Tab. 2)."*

*There is no citation to Herbstritt et al. (2023) because the sentence is now in the results. However, we refer to the memory effect they found in the Discussion.*

L379f (and elsewhere): I think it is not necessary to repeat the isotopic composition of the standards so often. Ideally, the outcome of your method should be independent of these values anyway.

*Changed.*

L382: Why did you stop at H7? It would also be important to confirm that the readings stay in that range.

*The measurements during this experiment took a long time, which meant that we were only able to carry out 7 repetitions within two days. As H5 and H6 were already close to the accurate range, we decided not to carry out any further measurements.*

L397ff: You advise to reuse bags but you did not show how the isotopic signature of unknown samples can be obtained in the foreseen application, i.e. remote sampling followed by lab-based analysis on a different day. In the storage and memory test you repeatedly flushed the reused bags with standard vapor until the readings were acceptable (after

irrelevantly short "storage" times). The proposed procedure (filling and emptying at least seven times (L400) and promptly measuring) is certainly not desirable (or feasible) when collecting unknown samples in remote locations. What would be the achievable sampling frequency in that case? And would that still be an advantage compared to direct in situ measurements performed with an analyzer that has been brought to the field?

*Thank you for your comment. We understand that with the explanations and results presented in the recent form, an unrestricted recommendation for reuse cannot be made. By splitting the October/February measurements with the additional explanation that rinsed and reused bags were used in February, we can currently only recommend this method for measurements in a narrow natural abundance range (and following strict guidelines). We performed an additional experiment (see comment above) to be able to make a statement about samples with larger differences in isotopic signature and discussed the results.*

*But to answer your questions for possible future experiments: Filling the sample bags ten times with the target sample in the field and then emptying them would make the system more complex, as one pump would be needed for filling and one for emptying. However, if a system were built for each sample bag that automatically fills (~15 minutes) and empties (~1 minute) the bags and collects the samples at the same time (you would need as many pumping systems as you have samples), such sampling could be done in about 3 hours with a theoretically unlimited number of samples.*

L400: With what and for how long should re-used bags be filled? I am sure this has in impact on feasible sample storage time. Can you also comment on a quantitative relationship between the ranges in isotopic compositions of previous samples and the necessary number of pre-sampling filling cycles?

*The bags were rinsed with dry air. These questions should now be clarified. We now explain it in the method section (2.4.4) and the "Handling Recommendations" (supplement).*

L404: Did you compare in situ measurements and bag measurement only during two or during all 18 campaigns? If two, then how were conditions different, especially regarding elapsed time between sampling and measurement and relative to the sample storage time tested in the combined experiment? Please specify in the method section.

*We have now rewritten this section with a separation of experiments and results/discussion for better understanding.*

- *Yes, only two of the 18 campaigns compared in situ and bag measurements.*
- *In the first campaign, we first measured in situ and then the bags immediately after filling (resulting in a direct/bag measurement in ~30 minutes).*
- *In the second campaign, we measured in situ and filled the bags. The bags were then measured in the lab within 24 hours after filling.*

L407: To make life easier for potential users of your method, please specify "filling errors". In addition, how did you identify condensation? Where did you see it?

*We have now added "Handling Recommendations" to the supplement for further details. Regarding condensation, we once measured a bag at a temperature that was too low (the AC flow was directed toward the bag), resulting in a small condensation peak during the bag measurement. Since we could not be sure that there was no effect on the rest of the sample, we*

*discarded this bag. Condensation during bags filling should be avoided by flushing the soil probes in the field with dry air prior to the measurement.*

L409: This is important and should appear in the method section already: What did you use for rinsing the bags and where was this step performed? Standard-derived vapor in the lab or the to-be-collected, unknown sample vapor in the field? If the latter, what was the required per-sample time required for this step? 10 x 15 min = 150 min?

*This statement has been moved to the method section (2.4.4) for better understanding. Also, the experiments are now better explained and discussed.*

*It now reads: „To exclude any memory effects, as we saw in experiment III, the reused bags were rinsed 10 times with dry air (approx. 10 x 10 min)."*

L432: On what kind of analyzers do co-extracted organic compounds interfere with water stable isotope measurements?

*Laser based cavity ring down spectrometer like the CRDS we used (Picarro 2310-i).*

*It now reads: "The accuracy of CVE can vary greatly for soil samples and is associated with co-extraction of organic compounds, significantly interfering with the isotopic quantification using CRDS (Orlowski et al., 2016b)."*

L444: After what?

*Deleted.*

L446: Please specify "wide"

*It now reads: "Measurements of soil water isotope profiles over the full season (Fig. 7) revealed a wide range of isotopic signatures with 2.1 ‰ to -15.2 ‰ for $\delta^{18}O$ and 12.9 ‰ to -98.5 ‰ for $\delta^2H$."*

L447: The period needs to be specified.

*It now reads: "The isotopic signature of precipitation is represented by the local meteoric water line (LMWL), shown here for the period of September 2021 to September 2023."*

L455: For additional plausibility, can you compare the nature of the scatter, e.g., by comparing the linearity ($R^2$) of the dataset, with that of precipitation data and other datasets of soil water isotopes? Is there a difference in linearity between the two campaigns with field-access to the analyzer and the other 16 without (if that was the difference)? How were standards produced and treated in these two different cases? How many validation standards were co-measured and what was their precision and accuracy?

*We have changed the graphic to better show the different campaigns and have added a more detailed comparison/explanation of the different depth and seasonal development (Figure S1). Three laboratory standards were bagged and treated in the same manner as the samples.*

L458: transpiration rather does not cause enrichment. Evaporation does. Please change "evapotranspiration" to "evaporative"

*Changed.*

L462f: Where do I find the seasonal variability you are referring to?

*Will be changed. It now reads: "Overall, our findings from the field trial suggest a good agreement with GPM probe and bag-based soil water isotope measurements with the LMWL and are plausible in terms of seasonal variability (see Fig. 6c; e.g. compare offsets between cryogenically extracted bulk soil water isotope measurements and LMWL; e.g. Zhao and Wang, 2021)."*

L465f: This seems to be a bit off. Usually, the lower boundary of the plow layer is around 20 cm, not 45 cm. Was it different in your case? Can you also comment on the large range of isotope values observed for 150 cm depth (yellow symbols in Fig. 6)? I would expect to see a less pronounced variation at that depth.

*It's correct that the lower boundary of the plow layer is typically located around 20 cm but it depends on the soil conditions during plowing (high soil water contents can lead deeper plowing). We actually expected the lower plow boundary to be 20 cm and consequently the deeper probes to be unaffected by tillage. Hence, the probes at 45 cm and 150 cm were not recovered and reinstalled before and after tillage. In comparison, we routinely remove/reinstall the soil probes in the upper layers (5cm and 15cm) during/after tillage. After discovering the very low vapor concentrations in the probes in 45 cm depths, we suspected damage to the probes due to the tillage. Personal communications with our field manager revealed, that the tillage was indeed deeper than 20 cm and likely resulted in a compaction of the soil down to the 45 cm probes. We have repeatedly tried to measure these probes and could measure some of them in a vapor concentration matching the vapor saturation at the given temperature. Those measurements were deemed likely to be valid and were included in the manuscript.*

L468: Why does soil compaction flaw the measurements? In situ measurements have been conducted successfully in boreholes of (I would say: rather compact) trees by one of the co-authors. So why wouldn't they work in compacted soil? And why would that be an issue at 45 cm but not at 150 cm depth?

*See comment above. (The compacted soil is not the problem per se, only the fact that the probes in 45 cm were installed before tillage i.e. they were in the soil when the compaction occurred, which is the typical handling of sensors in many agricultural studies, e.g. only de-install sensors that are above the manipulation depth)*

L475: I think, "appropriate" is inappropriate here. You did not test the effect on samples stored in reused bags for more than 1 hour. (Or you forgot to mention that.) Consequently, I do not see how reliable measurements of unknown samples stored for typical time periods in reused bags can be performed based on the findings of this study.

*This statement should be clear with the rewritten method section, the separation of results / discussion and considering the field experiment and the additional experiment.*

L476: rinsing with dry air does not match the procedure described in the combined memory and storage experiment. Please explain (before the conclusion), why rinsing with dry air – previously suspected to increase scatter – does (or should do) the same trick that flushing with moist air does.

*This statement should be clear with the rewritten method section, the separation of results / discussion and considering the field experiment and the additional experiment.*

L485: are these numbers based on two or on 18 campaigns?

*It now reads: "Through the conducted field experiment (two campaigns with CRDS and bag measurements), we were able to show that the bags could be used in our case with an accuracy of 0.23 ± 0.84 δ18O [‰] and 0.94 ± 2.69 δ2H [‰], which allows a wide applicability."*

L490: Not "can" but "will"

*Changed.*

S1: AMT is a European Journal. I suggest using the metric system and SI units.

*Changed to SI units.*

S2 & S3: What depths are you referring to? Weren't these measurements performed on standard vapor sampled in the lab?

*"Depth" will be deleted. It now reads: "Differences during the storage experiment for M22 and L22 for each storage duration…"*

---

## Author Response (AR2)

Dear Christof Janssen,

Thank you for your time and effort as an editor on our manuscript entitled "Simple water vapor sampling for stable isotope analysis using affordable valves and bags". Attached please find our revised manuscript to be considered for publication.

This work has been previously submitted to AMT, with "major/minor revisions" being suggested to apply to our manuscript. The main recommendations from you and the two reviewers were to better present the results, limitations, and restrictions of our approach throughout the manuscript, especially in the abstract and discussion.

We have substantially revised our manuscript according to these suggestions, taking into account all criticisms and comments and we believe the implemented revisions improved the manuscript significantly.

Sincerely,

Adrian Dahlmann

Dear Authors,

Thank you for submitting your manuscript to EGUsphere/AMT. As evident from the referee reports, your article merits publication in AMT. However, and this has been also indicated in one of the reports, some of the limitations of the proposed method are not clearly stated and the interpretation of the provided data is not fully convincing. Therefore, the current presentation is not yet fully in line with AMT standards and still requires improvement before it can be published. I thus decide that the paper requires a major revision (with review). In preparing the revised version, you should

1/ Consider the recommendations of both of the referees.

1/ All the comments made by the reviewers are now included in the revised manuscript.

2/ Improve the abstract to better reflect the findings (& their limits) of your article.

2/ We have now added more information to the abstract to clarify the results, limitations and restrictions of our method:

- Information about the use of new / reused bags were added to the result description:
   L20: "The storage experiment with new bags demonstrated the ability to store water vapor samples for up to seven days while maintaining acceptable results for δ2H, and acceptable to questionable results for δ18O. The memory experiment using new bags revealed that the influence of previous samples increased with duration of storage."
- The recommended use of similar samples was added:
   L27: "The reuse experiment confirmed that the bags can be filled repeatedly, provided they are used for similar sample lines and rinsed ten times with dry air."
- We point out that storing was tested mainly for 24 hours:
  L30: "Storing beyond 24 hours needs further investigation but appears promising.
  With new gas bags up to 24 hours of storage, we found accuracy of 0.2 ± 0.9 ‰ for δ18O and 0.7 ± 2.3 ‰ for δ2H. When the bags were reused and stored up to 24 hours, they yielded accuracy of 0.1 ± 0.8 ‰ for δ18O and 1.4 ± 3.3 ‰ for δ2H."
- The limitations are now highlighted: *L33: "The proposed system is simple, cost-efficient, and versatile for both lab and field applications, however, case-specific testing is necessary given the remaining uncertainties."*

3/ Do the same for the discussion and conclusions. I am worried about two particular issues here:

- First, the L22 results in experiment I seem to be completely ignored and are not mentioned in the discussion. In this test, only 2 measurements out of all 15 measurement points are in the 'accurate range'. These two points belong to the 7-day storage time. One interpretation is that this indicates that there is no clear temporal trend — and this seems to be the chosen route. However, another possible interpretation is that due to some unknown reason low  $\delta$ -values cannot be measured reliably by the bag method, which would invalidate your conclusions. It is likely that the Day 1 and Day 3 measurements suffer from contamination (a straight line through M22 and the lowest Day 1 measurement shows that all the incriminated (z>=2) day 1 and day 3 measurements are aligned along an isotopic mixing line), whose origin in the absence of further details is difficult to explain. For this reason, it is imperative that there is a discussion/evaluation of these particular results in the discussion section and that the impact on the validity of the method is discussed.

First, we explained and discussed the results in more detail. For this purpose, we have divided the discussion into two subsections, "4.1 Comparison to previous developments to store and measure water vapor " and "4.2 Limitations, future perspectives and cost classification". In particular, we will highlight the experimental results and provide an explanation of our findings and the uncertainties that have arisen.

- Second, the discussion section starts by comparing the current results with previous studies and claims a comparable performance (L388 and following "Our results are generally comparable in accuracy to previous studies of water vapor storage. For example, the Soil Water Isotope Storage System (SWISS) introduced by Havranek et al. (2020) showed a high accuracy during a 30-day storage period in a laboratory experiment ( $\pm 0.5 \% \delta 180$  and  $\pm 2.4$ ‰  $\delta 2H$ ). This result was followed by several experiments, which showed an actual precision of 0.9 ‰ and 3.7‰ for  $\delta 180$  and  $\delta 2H$  in field applications with a storage time of 14 days (Havranek et al., 2023)." This is misleading, because the current study has only looked at storage times of not more than 7 days. How can these be compared to maximum storage times of 14 and 30 days of the two other studies? Moreover, the majority of the Havranek 2020 samples (9 for 180 and 10 for D) out of all 12 samples (6 overnight samples, 3 24-day storage samples and 3 30-day storage samples) with a target at  $\delta D = -122$ ‰ and  $\delta 180 = -16.4$ ‰ fall in the z <= 2 range (as expected from basic statistics). This is certainly not consistent and comparable with your results on the somewhat similar isotopic target L22 (with  $\delta D = -148.1$  % and  $\delta 180 = -19.9$  %) presented in Fig. 3c, where only a minority of samples is within the  $z \le 2$  range.

Second, we agree that our results are not directly comparable to other methods such as Havreneck et al. (2020). We have now removed the first sentence ("Our results are generally comparable in accuracy to previous studies of water vapor storage.") to avoid misunderstanding. The brief introduction of the SWISS system by Havreneck et al. (2020) is still part of the discussion, but we now state that the accuracy is higher with their system and we strongly believe that it is necessary and fair to compare/classify similar methods when all information about differences is given:

L393: "In general, it is difficult to compare the few different approaches to water vapor sampling for isotopic analysis because they vary in complexity and application (e.g., storage time or price per sample). However, our results for reused bag samples stored up to 24 hours are generally comparable in accuracy to previous studies of water vapor storage. For example, the Soil Water Isotope Storage System (SWISS) introduced by Havranek et al. (2020) showed a higher precision during a 30-day storage period in a laboratory experiment  $(\pm 0.5 \% \delta^{18}O \text{ and } \pm 2.4 \% \delta^{2}H)$ . ..."

In addition, we have now added the information about our recommended storage time of up to 24 hours throughout the manuscript, and regarding Havreneck et al. (2020), we highlight the fact that glass methods may be superior for long-term storage.

4/ Explain and justify why memory tests with storage (experiment III) have been restricted to measurement of the H standard after passing the L standard. In both previous experimental tests (storage (experiment I) and memory (experiment II)), L and H standards have been treated interchangeably and it has been shown that for some unknown reason, measurement of L is more critical and prone to contamination than measurement of M or H. Including the reverse measurement should provide a more realistic qualitative assessment of your approach and the results could have an impact on the required number of purges when utilising used bags.

Thank you for your comment. We understand your concern about the light standard and have now explained and discussed the issue throughout the manuscript. Below is some information to answer your questions:

- Experiment I only tested storage with new bags for up to 7 days, while Experiment III showed the memory effect with two very different standards (overnight storage with

the initial standard here resulted in acceptable values). Therefore, the results of experiment I, with a higher uncertainty for  $\delta^{18}$ O compared to predominantly acceptable values for  $\delta^{2}$ H, cannot be related to experiment III.

- The larger uncertainty of the light standard of experiment II was indeed insufficiently discussed. Unfortunately, the explanation was deleted during the first round of revisions as part of the separation of the discussion. We have now added information that the stronger memory effect was caused by different handling of the samples (approx. 45 minutes of storage). No other results indicate that the heavy standard is easier to handle than the light standard.
- Experiment III was only carried out in one direction, as we first wanted to quantify the memory effect and in particular to show how long it could affect the subsequent samples.

However, by revising the manuscript, we have now clarified that the method can only be used for the range of isotopic signatures we tested. Further investigations for other ranges and directions, such as the experiment you recommended with the combined memory/storage from heavy to light standard, should be carried out. Unfortunately, we are unable to do this at this time due to construction in our labs.

5/ Introduce and use a consistent terminology around measurement uncertainty. It appears that you are following actual recommendations from BIPM as stated in the VIM ("JCGM 200:2012, International vocabulary of metrology – Basic and general concepts and associated terms (VIM),

3rd edition", electronic link provided below). Sometimes in the manuscript, however, the accuracy is specified which is contrary to these recommendations and it is particularly confusing to see that the accuracy is once defined as a range ( $\pm 0.5 \%$ , L391 for example), and once as a measurement result with an associated uncertainty range  $0.25 \pm 0.41\%$  (e.g., L408). I strongly recommend to use consistent terminology and notation (in preference in line with the recommendations from BIPM and ISO) and stick to that throughout the manuscript. Following VIM, it would be more correct to talk about a trueness range instead of the accurate range for z <= 2.

We have carefully revised the manuscript according to BIPM/ISO terminology and notation from and have redefined the z-score definitions, e.g.:

- L31: "With new gas bags up to 24 hours of storage, we found accuracy of  $0.2 \pm 0.9$  ‰ for  $\delta^{18}O$  and  $0.7 \pm 2.3$  ‰ for  $\delta^{2}H$ ."

- L290: "A z-score < 2 represents an acceptable range, a z-score between 2 and 5 describes the questionable range, and a z-score > 5 representing an unacceptable range (Wassenaar et al., 2012; Orlowski et al., 2016a)."
- L310: "Consequently, z-scores were either within the acceptable range or close to it, again with no trend of decreasing accuracy over storage time."

**Minor issues:**

- L291: There doesn't seem to be a gradient here. Replace by 'range'.

**Changed.**

- L204-211: The experimental description could be improved by directly indicating the number of bags at the first instance and the number of repeats (3), ie '5 gas bags' in L206 and 'twice with the opposite standard' in L208. The last phrase can then be deleted.

Thank you for your comment. We now added *"utilizing five newly prepared bags per standard."* in L215 and the number of repeats *"We repeated the process three times (fill, measure, empty) with the opposite standard..."* in L218.

- L234/235: add space in front of 'cm'

**Changed.**

- L321: remove 'again', as it is the first time that the memory effect disappears.

**Changed.**

- L503: The doi of Havranek 2020 leads to a paper of Lee, Lee and Yoo entitled "Analysis of ceramides in cosmetics by reversed-phase liquid chromatography/electrospray ionization mass spectrometry with collision-induced dissociation". There is likely just a digit missing in the doi number.

**Indeed! There was a 3 missing at the end. Thank you!**

- Fig S2: Why is the z-score scale for  $\delta 180$  flipped as compared to the direct  $\delta 180$  scale? The data points seem to have undergone a reflection at the x = 0 axis when going from the left to the right box — which should not be the case.

Thank you for your comment. We have changed the x-axis.

**References:**

VIM https://www.bipm.org/documents/20126/2071204/JCGM\_200\_2012.pdf/f0e1ad45- d337-bbeb-53a6-15fe649d0ff1

**Report #1**

**General comments:**

I appreciate the effort the authors have put into revising and improving the manuscript. However, I still have some concerns regarding the interpretation of the presented findings. In general, it appears to me that the obtained data quality requires certain circumstances to be achievable. Specifically, rather short sample storage times and a strict "identical bags for identical probes" procedure were necessary to achieve the obtained data quality. Further, the seven-day storage test was not performed on unknown samples in reused bags which would make the method more applicable for potential users. Also, you state that your method is somewhat safe for samples in the natural abundance range but the memory test (3.3) was performed on standards representing this range and required numerous refills in order to get accurate isotope readings. Therefore, I think these restrictions need to be stated in the abstract already, not only hidden deep inside the manuscript.

You showed that flushing reused bags ten times with dry air yielded bag data in agreement with in situ measurements after one day of storage. Notwithstanding the fact that this procedure (or the number of necessary flushes) was not tested prior to field application, I think this is the main argument supporting your story. All other data are either not representative or show the limitations of your method and the importance of sticking to some narrow requirements that make your method less universal or practical. I strongly suggest to make this (short storage time, strict "identical bags for identical probes" procedure) clearer throughout the manuscript.

Thank you for reviewing our manuscript. We have now tried to present the findings, limitations, and restrictions of our method more clearly in the abstract (see the editor's general comment 2/), but also to discuss them in more detail (see the editor's general comments for specific citations).

In particular, we clearly state throughout the manuscript that storage with the given experiments can only be recommended for up to 24 hours and ideally for similar samples.

Regarding experiment III, we showed the memory effect only after one day of storage of two very different standards (light to heavy) without flushing of the bags. Here we have tried to quantify the memory effect without trying to remove it. But you are right, the additional experiment S2 revealed that memory effects still can occur even with 10 flushes. We have added this restriction to the abstract/discussion and note that this method should be further tested for different experimental designs.

**Specific comments:**

L10: insert "of soil or plant water isotopes" after "measurements" as you are talking about matrix-bound water that requires extraction.

**Changed.**

L11: consider replacing "semi-permeable" by "vapor-permeable" or "hydrophobic"

Changed to "gas-permeable membranes" for consistency.

L21: I find this statement somewhat misleading. "seven days" seems to suggest that unknown samples can be stored for such a long time. In reality, the performance was a function of isotope values with the standard close to ambient air performing best.

We understand that this could be confusing. We now changed it to:

L20: "The storage experiment with new bags demonstrated the ability to store water vapor samples for up to seven days while maintaining mostly acceptable trueness for  $\delta^2 H$ , and acceptable to questionable trueness for  $\delta^{18}O$ ."

In addition, we have added the storage time to the results at the end of the abstract to avoid misunderstandings:

L30:,,Storing beyond 24 hours needs further investigation but appears promising. With new gas bags up to 24 hours of storage, we found accuracy of  $0.2 \pm 0.9$  %, respectively, for  $\delta^{18}O$  and  $0.7 \pm 2.3$  % for  $\delta^{2}H$ . When the bags were reused and stored up to 24 hours, they yielded accuracy of  $0.1 \pm 0.8$  % for  $\delta^{18}O$  and  $1.4 \pm 3.3$  % for  $\delta^{2}H$ ."

L23: replace "samples" by "bags"

**Changed.**

L28: do you mean "replicate measurements"? "Repeated measurements" seems to suggest multiple analyses of the same bag(s).

Yes, we have changed it to "replicate measurements" now.

L31: "cost-effective" or rather "cost-efficient"?

Changed.

L49: To my knowledge, Marshall et al (2020) should not be on this list as they did not use membranes when drilling holes through tree stems.

Thank you for your comment. It is correct that Marshall et al. (2020) didn't used gas permeable membranes. We now cite the Paper "Xylem water in riparian willow trees (*Salix alba*) reveals shallow sources of root water uptake by in situ monitoring of stable water isotopes" by Landgraf et al. (2022). Here, the borehole equilibration method was used with gas permeable membranes.

L52: The (potential) difference in costs is mainly the power source as for subsequent, labbased measurements a costly analyzer is needed as well.

It is true that an expensive analyzer is also needed, but not in the field. With our method, one lab analyzer could be used to measure numerous field sites. It now reads:

L53: "However, direct, continuous in situ field setups are very cost-intensive, technically challenging and require a permanent power supply in the field as well as strong expertise to maintain. Moreover, direct in situ field setups require full-time operation of one laser spectrometer (e.g. a CRDS) each, whereas a vapor storage method could be operated with one CRDS for several field setups."

L53: "strong expertise" is also required (or at least favourable) for lab-based analyses. - Hail to the lab personnel! ;)

**You are absolutely right!**

L60: temperature stability is less important than consistently exceeding the sampling temperature during analysis

We now changed this sentence to clarify this statement:

L63: "The advantages of these methods include the ability to quickly measure stored samples at elevated temperatures relative to the source in a temperature-stable laboratory environment. "

L109: I think you mean "consistent" when you say "accurate". Also, in this paragraph can you explain how you get from raw analyzer readings to VSMOW-referenced vapor values before you calculate liquid water values using Majoube`s equation?

"Accurate" was changed to "consistent".

Regarding the calibration, we added a new section in the end:

L127: "In laboratory experiments, calibration was performed by measuring the described glass bottles before the start of the measurement and the used standard during and after the experiment for drift correction. In field experiments, the standards covering the expected sampled isotopic range were filled into bags and treated similarly to the samples. Calibration was then performed."

L133: Please insert "nominal" before "capacity" as 90% is the actual, usable filling capacity.

**Changed.**

L148: Please insert "Flow at" before "the open split".

**Changed.**

L219: It still puzzles me that you emptied the bags over night instead of testing the effect of this scenario which is probably representative for many occasion when analyses are not performed on the same day as sampling.

At this stage of our research, we only wanted to quantify the effect of the first standard stored in the bag. If we had left the samples filled over the break, we would not have been able to guarantee this.

The following experiment IV was conducted for potential effects that could occur overnight.

L245: How were "concurrently measured" raw vapour concentration readings of the Picarro calibrated?

The calibration is now explained in the end of section 2.1.

L406: Delete "known" as their method does not require knowledge of ambient vapour isotope values.

Thank you for your comment. It now reads:

L413: "To circumvent these memory effects, they explored preconditioning of the bags with moist, isotopically homogeneous air sample where the goal was not to eliminate the memory effect, but to make it predictable and remove it."

L415: Did Herbstritt et al. (2023) report leakages or why is this considered an important difference to their approach?

No, they did not report any leaks, but we wanted to emphasize that we are building an easyto-use connection. We have now rewritten this sentence to focus on the simplification of gas transfer.

L422: "Second, we have modified the valve inlets to the bags in a way that simplified gas transfers and may reduce leakage."

L429: By saying "we know the previous sample signature", do you imply that bags can ONLY be reused for identical samples/probes? What if I have a limited set of bags and want to use it on different locations before repeat sampling at the location visited first? If this does not work – not even with flushing dry air ten times - you should say so.

We now clearly state throughout the manuscript that sampling can only be recommended for the given storage time of up to 24 hours and is also limited by the isotopic signature of consecutive samples. In particular, we have added the final sentence in this section:

L439: "Concluding, our results suggest comparable accuracy to other methods for 24 hours, but the accuracy of long-term storage and high isotopic differences for consecutive samples should be further tested."

L438: Evaporation does not cause scatter "around" the LMWL but rather below as evaporation lines have slopes significantly smaller than the one of the LMWL.

We now changed it to:

L447: "This results in a wide range of isotopic signatures throughout the complete cultivation season, as can be seen in the smaller slope compared to the LMWL in the upper soil layer (Fig. 7)."

L455: "relatively low" is misleading as only on study (Havranek et al., 2020) reported higher per-container costs. Or do you mean "relative to overall expenses of a typical field campaign"?

Yes, it now reads:

L504: "The cost of the commercial gas bags we used was relatively low compared to the total cost of a typical field campaign."

L467: "cost-effective" of "cost-efficient"?

Changed.

L472f: As the method should ideally be applicable for unknown samples, it is quite a limitation that differences between consecutive samples should be small. Reference to the natural abundance range is misleading in this context as Experiment III (using standards within that range) showed that numerous refills were required before good data were recorded.

See general comments above.

**Report #2**

Thank you for your effort in the revision of the manuscript. However, I still have a few minor comments.

**Specific comments:**

L. 49: Marshall et al. didn't use gas permeable membranes in their borehole method.

Thank you for your comment. It is correct that Marshall et al. (2020) didn't used gas permeable membranes. We now cite the Paper "Xylem water in riparian willow trees (*Salix alba*) reveals shallow sources of root water uptake by in situ monitoring of stable water isotopes" by Landgraf et al. 2022. Here, the borehole equilibration method was used with gas permeable membranes.

L. 412 ff.: The first three points are the differences, whereas the fourth point is in agreement with the paper of Herbstritt et al. (2023). I therefore recommend rephrasing the beginning of the sentence in L. 417, e.g., 'In agreement with Herbstritt et al., (2023), we have identified..."

We totally agree and changed the beginning of the sentence:

L425: "Aside from the differences, we likewise identified a time-dependent memory effect, which is consistent with the notion that some diffusion/adsorption process occurs over many hours within the walls of the bag, setting an isotopic signal that requires multiple flushes to remove."

L. 425 and L. 470: "...ten-times flushing with dry air (Fig. S2)..." I can't find any evidence in Fig. S2, why dry air flushing is more recommendable over moist air flushing. Fig. S2 seems to be about new bags vs. reused bags after dry air flushing. I don't see any dry vs. moist flushing data. Did you do this comparison? If not, you can't proof the statement in L. 470 and should delete it in the Conclusion section.

Thank you for your comment. It is correct that the additional experiment did not test a comparison between dry and moist air flushing. Based on your comment, we have removed the statement in the conclusion.

**Technical correction:**

L. 109: ...results up to 100...

Changed.

Figure 3: you could perhaps add 'M22' to 3b and 'L22' to 3c, according to the labeling of 3a.

Changed (also for Figure 4 for consistency).

Figure 5, caption: Please begin with the description of (a), then (b). Where is the 'arrow'?

Changed caption and added arrow.

L. 403: replace the ";" between 'et al.' and '2023' by a ","

Changed.

L. 459: delete 'our' after "...commercially available..."

Changed.

---

## Author Response (AR3)

**Public justification (visible to the public if the article is accepted and published):**

Dear Authors,

I am sorry for the somewhat edgy review process, but the presentation of your results needed some reworking, which I realised only in a late stage of the processing. Your actual revision has done that very well. So, thank you for taking into account all the previous editorial comments, which finally makes your manuscript publishable in AMT. Rereading the whole document, there are very few and minor, mostly technical corrections that need to be made before the manuscript will be published, however.

**Dear Christof Janssen,**

Thank you for your feedback and for guiding us through the revision process. We are pleased to hear that the manuscript is now considered publishable in AMT.

We have carefully addressed the remaining minor corrections as follows:

L. 31-33 Write (0.2 ± 0.9) ‰ etc, We found accuracy -> We found accuracies, they yielded accuracy of -> they yielded accuracies of Since ‰ is to be treated as a real number (see below), where standard algebraic laws should apply, the distributive rule must be respected and parentheses cannot be omitted, here and elsewhere in the manuscript.

The entire manuscript has been reworked with respect to your recommendation to rephrase "accuracy" to "accuracies" when both values ( $\delta$ 180 and  $\delta$ 2H) are mentioned (accuracies are now written as e.g. 0.7 ‰ ± 2.3  $\delta$ 2H). For isotopic values, see the comment below for L95/L271/L273.

L. 51 I could only find Kübert et al 2020 in the list of references. Please correct Kübert et al 2021

**Done.**

L. 560 : While the text has been corrected, the electronic link embedded in the pdf has not been updated and does not yet include the missing digit (3). It still directs to an article on cosmetics.

**Well, I did not know that was possible. I have now replaced the whole link.**

L95, L271, L273. The different definitions you are giving in eps 1 to 3 are conflicting due to the introduction of additional (and historical) factors of 1000 being included in some definitions but omitted in others. For example, if 1000 is included in the

definition of  $\delta$ -values (eq 1), then eq 2 would need to contain  $\delta$ /1000 instead of  $\delta$ . The same contradiction holds for the definition of  $\alpha$ + or ln  $\alpha$ + in equations 2 and 3. Once, a factor of 1000 is included (eq. 3), once it is not (eq 2). Metrological institutions commonly recommend to treat the percent and permil signs as mere numbers (and not units) with % = 0.01 and ‰ = 0.001. The recommended practice is thus to use all definitions without additional factors of 1000 and use the identity 1 = 1000 ‰ to express  $\delta$ -values in permil etc. See for example the IUPAC technical report, doi:10.1515/pac-2013-1023. If these recommendations are followed, the reference to Kübert et al., 2020 in line 263 should be deleted. Finally, the reference to Craig, 1961 in line 94 could also be deleted.

Many thanks for your recommendation. It is indeed correct that our equations 1-3 are not consistent regarding the  $\delta$ -notation in permil. We have now changed all of the equations to be correct for the use in permil instead of mere numbers. While we generally agree with your suggestion to amend this, we believe that the notation in permil is such a common practice in the stable isotope community that it would be potentially confusing for the main target group to not represent equations in delta notation.

L. 357. Add a full stop at the end of the figure caption.

We assume that you are referring to the empty line 357 and added a line break (punctuation sign was already there) at the end of the figure 5 caption to avoid the empty line.

L. 416. You probably intend to say "but to quantify and correct for it".

**Done.**

L. 439-441. Concluding, our results suggest comparable accuracy to other methods for 24 hours, -> In conclusion, our results show comparable accuracy to other methods for storage times of up to 24 hours,

I also suggest to write "require further investigation" instead of "should be further tested".

**Done.**

L. 508 - 513. There is a mix of tenses. It is best to write : "We have demonstrated that" and then switch to present tense for the remaining part of the phrase ("available bags meet the expected level, etc.").

**Done.**

With kind regards Christof Janssen